# Architectural and evolutionary features of TE-derived TSSs shape tissue-specific promoter activity in the human genome

Ya Zhang [1] ✉, Yahan Fan[2], Huang Wu[2] & Xiao-Ou Zhang [1] ✉

Transposable elements are abundant in the human genome and have been increasingly recognized as sources of alternative promoters. Yet, the extent of their transcriptional activity in human tissues and the features that govern their regulatory potential remain unclear. Here, we integrated high-resolution RAMPAGE data from 115 human biosamples to construct a comprehensive atlas of 26,056 transcription start sites derived from transposable elements. These sites contribute to tissue-specific gene expression, with a notable fraction originating from primate- and hominid-specific elements. Transposable element–derived transcription start sites exhibit focused, narrow-peak architectures enriched for TATA boxes and depleted of CpG islands. Phylogenetic analyses reveal a continuous gradient in promoter strength and transcriptional precision across transposable element subfamilies, with evolutionarily younger elements retaining intrinsic promoter motifs that drive focused and robust transcription, whereas older, more divergent elements exhibit broader initiation patterns and lower intrinsic activity. Together, these findings advance our understanding of how the evolution and preservation of promoter features shape the capacity of transposable elements to be exapted as functional promoters, potentially contributing to lineage-specific regulatory innovation in primates.

Transposable elements (TEs) are ubiquitous mobile genetic sequences that collectively constitute a substantial fraction of eukaryotic genomes and have served as pivotal driving forces shaping their architecture and function[1,2]. Approximately 45% of the human genome derives from multiple TE classes, including long interspersed nuclear elements (LINEs), short interspersed nuclear elements (SINEs), long terminal repeat (LTR) retrotransposons, and DNA transposons[3,4]. By virtue of their capacity to mobilize, these elements have been co-opted over evolutionary time to drive the emergence of novel regulatory modules, often catalyzing rapid innovation in gene expression programs[5]. To mitigate the potentially deleterious effects of

unchecked transposition, the host genome has evolved multilayered epigenetic defenses, in particular DNA methylation and repressive histone modifications, to silence the vast majority of TE copies in somatic cells[6,7]. Nevertheless, accumulating evidence indicates that many TEs escape these repressive marks and are repurposed as functional regulatory elements, actively involved in key biological processes such as embryogenesis[8,9], immune responses[10,11], aging[12], and tumorigenesis[13,14]. In particular, certain TE copies can function as alternative promoters of neighboring genes–initiating transcription within the TE sequence to yield TE-derived transcription start sites (TSSs)–thereby modulating host gene expression[9,14,15]. In our previous

[1]Shanghai Key Laboratory of Maternal and Fetal Medicine, Clinical and Translational Research Center of Shanghai First Maternity and Infant Hospital, Frontier Science Center for Stem Cell Research, School of Life Sciences and Technology, Tongji University, Shanghai, China. [2]Department of Transfusion Medicine, Daping Hospital, State Key Laboratory of Trauma and Chemical Poisoning, Army Medical University, Chongqing, China. ✉e-mail: zhangaya1225@tongji.edu.cn; zhangxiaoou@tongji.edu.cn

work, we systematically annotated 5768 human TE-derived TSSs from 1205 high-quality RNA-seq datasets spanning multiple tissue and cell types, demonstrating that TE-derived promoters are a widespread phenomenon across diverse biological contexts[16].

Unlike autonomous TE transcripts initiated from canonical TE promoters, TE-derived TSSs represent co-opted regulatory elements that function as alternative promoters for nearby host genes[5]. These elements often harbor multiple transcription factor binding sites and core promoter motifs, enabling recruitment of the transcriptional machinery to initiate transcription of neighboring genes. For example, in lung cancer, an *AluJb* insertion upstream of the LIN28B oncogene serves as an alternative TSS, driving cancer-specific expression of LIN28B isoforms initiated within the *AluJb* element[14]. Similarly, during mouse preimplantation development, a mouse-specific *MT2B2*-derived TSS produces an N-terminally truncated Cdk2ap1 isoform that predominates in early embryos and enhances cellular proliferation[9]. Although TE-derived TSSs acting as alternative promoters have been well documented across diverse biological contexts, both the extent to which they contribute to total transcriptional output and the conditions under which they serve as the predominant TSS of a gene remain largely unexplored.

Beyond their context-specific roles, the evolutionary emergence of TE-derived TSSs remains poorly understood. In some cases, particularly within primate-specific ERVL-MaLR elements, the inserted LTR sequences already harbor core promoter motifs, enabling immediate transcriptional initiation upon insertion[17]. By contrast, other TE insertions may lack obvious promoter signals upon integration; cryptic motifs buried within their sequences can be exposed by alterations in the local chromatin environment, or point mutations may later generate de novo promoter activity[18,19]. As a result, nascent TE-derived TSSs are initially subject to purifying selection due to their potential to interfere with host gene expression, yet a subset resolves host–element conflicts and, through neutral drift or positive selection, becomes stably incorporated into existing regulatory networks[20,21]. Moreover, TE families differ widely in insertion age, indicating that both evolutionary timescale and genomic context crucially shape their promoter potential[9,22]. Importantly, the precise molecular mechanisms by which distinct TE families acquire promoter function, and the conditions under which they become the predominant TSSs of host genes, remain unknown.

In this study, we leveraged high-quality RAMPAGE (RNA Annotation and Mapping of Promoters for the Analysis of Gene Expression) data across 115 human biosamples to curate and characterize 26,056 TE-derived TSSs, covering 87 tissues and 28 cell types, among which testis showed the highest counts and greatest diversity. Functionally, TE-derived TSSs contribute to tissue-specific gene expression, with a notable fraction originating from primate- and hominid-specific elements. Unlike genomic TEs, which are generally depleted from promoter regions, TE-derived TSSs adopt focused, narrow-peak architectures marked by TATA boxes and low CpG density−hallmarks of tissue-specific Pol II promoters. Our phylogenetic analyses reveal that younger TE subfamilies retain intrinsic promoter elements driving precise, high-strength initiation. By contrast, older, more divergent families exhibit dispersed TSS positioning, attenuated intrinsic activity, and partial loss of structural integrity. These findings highlight how preserved core promoter features and evolutionary divergence jointly shape the potential of TEs to be exapted as functional, lineage-specific promoters.

## Results

### Genome-wide identification of TE-derived TSS with RAMPAGE data

RAMPAGE[23] employs paired-end sequencing of 5′-capped RNAs to map TSSs at single-nucleotide resolution and unambiguously link each TSS to its parent transcript. We previously curated high-quality RAMPAGE datasets from 115 biosamples (Supplementary Data 1) to accurately profile promoters of autonomous TE transcripts[24] and protein-coding genes[25]. To systematically identify TE-derived TSSs, we built a three-step pipeline (Fig. 1A) extending our prior methods. First, we applied our previously developed entropy (E) metric[24] during peak calling to distinguish true TSS peaks from background noise in repetitive loci (Supplementary Fig. 1A, left), and retained peaks overlapping annotated transposable elements as TE-overlapping TSSs. Next, we filtered these TE-overlapping TSSs by requiring that their paired reads extend into exonic regions of known or de novo assembled transcripts in the matched RNA-seq dataset (Supplementary Fig. 1A, right), ensuring transcriptional connectivity to host genes. Unlike short, unspliced autonomous TE transcripts[24], transcripts initiated from TE-derived TSSs are typically spliced into exons, producing RAMPAGE read pairs that span substantially larger genomic distances (Supplementary Fig. 1B). Because RAMPAGE library fragments are typically shorter than 1 kb, read pairs spanning longer distances likely represent spliced transcripts rather than autonomous TE transcripts. We therefore required a minimum genomic span exceeding this fragment length for candidate read pairs. In addition, since over 75% of RAMPAGE read pairs from expressed genes or host genes with annotated TE-derived TSSs exhibited exon-assignment fractions above 0.5 (Supplementary Fig. 1C), we set this value as the threshold to ensure reliable gene association. Together, these criteria effectively enrich bona fide TE-derived TSSs while filtering out autonomous TE transcription.

As illustrated in Fig. 1B, we identified an unannotated alternative TSS for CFAP210 embedded within an intronic *LTR52* element; this site exhibits strong H3K4me3 and DNase I hypersensitivity, and its read pairs directly connect the *LTR52*-derived TSS to downstream CFAP210 exons. When applied genome-wide, we observed that, unlike other TE-overlapping TSSs located far from genes, most TE-derived TSSs map within 1 kb of canonical promoters (Supplementary Fig. 1D). We therefore designate all TE-derived TSSs within this 1 kb window as high-confidence candidates (see Methods).

To evaluate the accuracy and comprehensiveness of TE-derived TSSs identified by our RAMPAGE pipeline, we performed multiple orthogonal validation approaches. First, we confirmed that TE-derived TSSs recapitulate key promoter features. They are strongly enriched for the initiator (Inr) motif at the +1 position (Fig. 1C) and exhibit significantly higher intrinsic transcriptional activity in SuRE assays[26] than other TE-overlapping TSSs (Fig. 1D), although their activity is slightly lower than that of canonical TSSs. Luciferase reporter assays, using four randomly selected TE-derived TSSs cloned upstream of a luciferase reporter, showed that all TSSs drove activity significantly above the empty vector control and higher than two previously reported TE-derived TSSs[27] (Supplementary Fig. 2A). In contrast, antisense insertion of the same TE-derived sequences markedly reduced luciferase activity, indicating strand-specific promoter activity (Fig. 1E). Expressed TE-derived TSSs also showed markedly stronger DNase I hypersensitivity, RNA Pol II occupancy, and H3K4me3 enrichment at their promoter regions compared to other TE-overlapping TSSs (Supplementary Fig. 2B), consistent with open chromatin and active transcription initiation.

To assess whether TE-derived TSSs give rise to processed transcripts rather than autonomous TE transcription, we examined sequence patterns within the 1 kb downstream region of each TE-derived TSS. These regions recapitulate canonical promoter features characteristic of mRNA TSSs (Fig. 1F), including enrichment of predicted 5′ splice-site motifs and depletion of polyadenylation signals[28], suggesting that TE-derived TSSs are likely to initiate processed transcripts. Consistently, a significantly higher proportion of TE-derived TSSs were supported by splice-junction-spanning RAMPAGE read pairs compared with other TE-overlapping TSSs (93.8% versus 1.4%; one-sided Wilcoxon rank-sum test *P*-value $< 2.2 \times 10^{-16}$; Fig. 1G). Independent de novo assembly of RAMPAGE reads further confirmed that TE-derived TSSs generate genuine, multi-exonic transcripts, whereas

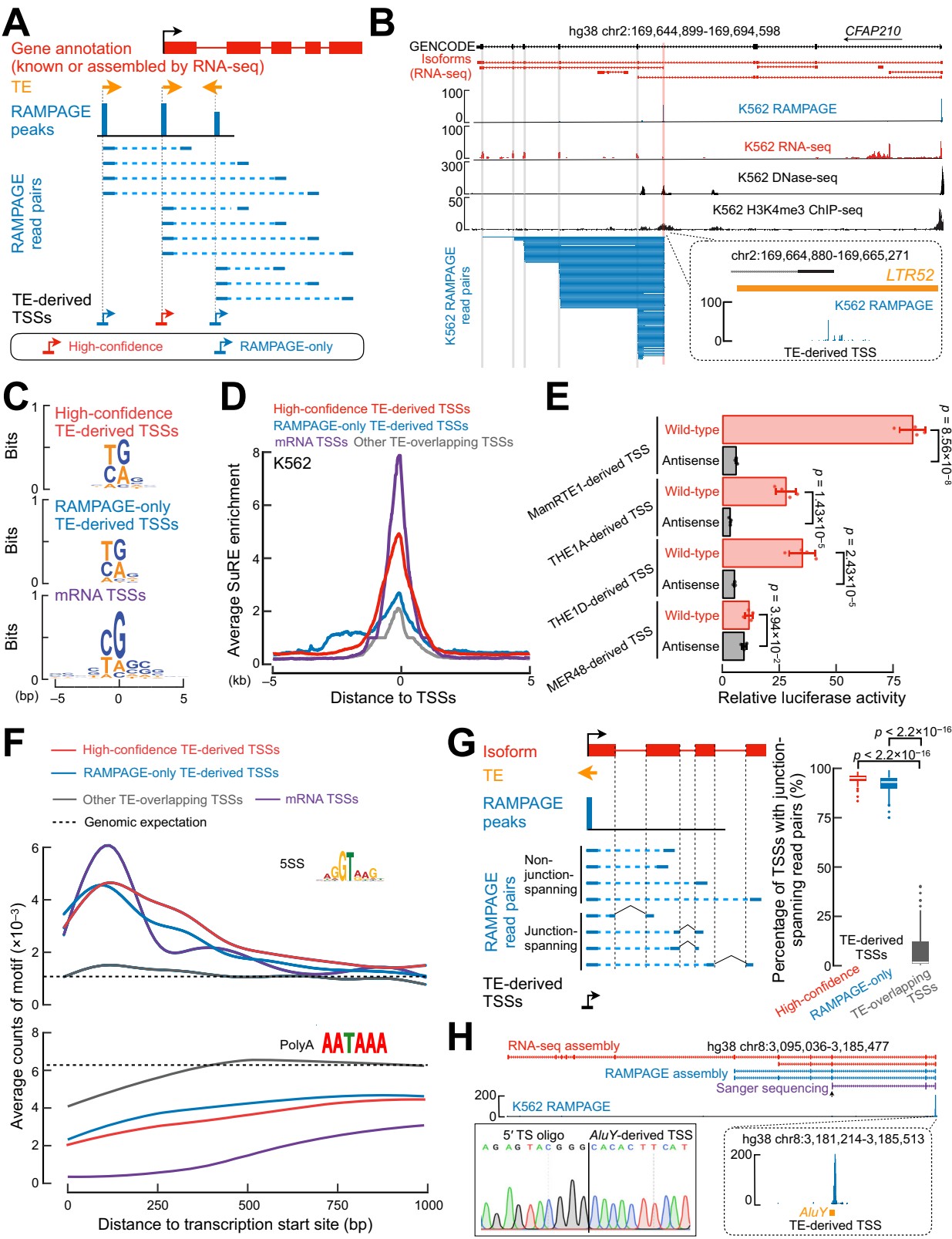

assemblies from other TE-overlapping sites remained short and predominantly mono-exonic (Supplementary Figs. 2C, D). Furthermore, 5′ rapid amplification of cDNA ends (5′ RACE) and Sanger sequencing of four randomly selected, previously unannotated TE-derived TSSs in K562 cells confirmed the transcription initiation events identified by our pipeline (Fig. 1H; Supplementary Fig. 2E).

Having confirmed both functional and transcript-level evidence, we next evaluated the reproducibility of TE-derived TSS detection across 50 biosamples with biological replicates. The median Pearson correlation coefficient of TE-derived TSS signal was 0.90 (Supplementary Fig. 2F), demonstrating the strong reproducibility of our RAMPAGE calls.

**Fig. 1 | Genome-wide identification of TE-derived TSSs using RAMPAGE data.**
**A** Schematic of TE-derived TSS identification pipeline using RAMPAGE data. TE-derived TSSs were defined by RAMPAGE peaks overlapping TEs and linked to downstream exons by RAMPAGE read pairs. High-confidence TE-derived TSSs (red arrows) were annotated by GENCODE or RefSeq or supported by RNA-seq–assembled transcripts from the same biosample, whereas the remaining TE-derived TSSs were deemed RAMPAGE-only (blue arrows). **B** Visualization of one TE-derived TSS for CFAP210 in K562 cells. A RAMPAGE peak (red vertical bar) marks the TE-derived TSS with RAMPAGE read pairs linking this TSS to downstream exonic positions (gray vertical bars). RNA-seq and H3K4me3 ChIP-seq datasets further confirm the expression of this TE-derived TSS. **C** Sequence logos of high-confidence TE-derived TSSs (top), RAMPAGE-only TE-derived TSSs (middle), and known TSSs (bottom). **D** Average SuRE enrichment with respect to high-confidence TE-derived TSSs (red), RAMPAGE-only TE-derived TSSs (blue), other TE-overlapping TSSs (gray), and TSSs of protein-coding genes. **E** Luciferase reporter assays measuring promoter activity of sense (wild-type) and antisense TE-derived TSS sequences. Data are shown as mean ± SEM from four independent

experiments. Luciferase activity was first normalized to Renilla luciferase and then expressed relative to the empty vector. Statistical significance was determined using the one-sided Student's *t* test. **F** Frequencies of 5′ splice site motif (SSS, top) and the polyadenylation signal hexamer (polyA, bottom) in the 1000-nt region downstream of high-confidence TE-derived TSSs (red lines), RAMPAGE-only TE-derived TSSs (blue lines), a randomly selected set of mRNA TSSs from GENCODE (purple lines), and other TE-overlapping TSSs (gray lines). Two dashed horizontal lines indicate the expected frequencies in the random genomic background.
**G** Percentage of TSSs supported by splice-junction-spanning RAMPAGE read pairs. Each boxplot summarizes data from 115 biosamples, with the center line indicating the median, the box representing the interquartile range, and the whiskers extending to the most extreme values within 1.5× the interquartile range. One-sided Wilcoxon rank-sum test *P*-values are shown. **H** 5′ RACE and Sanger sequencing transcription from a previously unannotated TE-derived TSS. RNA-seq and RAMPAGE-assembled isoforms are shown. The gene specific primer is indicated by an arrow.

To assess methodological robustness, we compared our results with TEProF2[29], an RNA-seq–based approach that identifies TE-derived TSSs through transcript assembly and read assignment. In representative samples including GM12878, K562, heart, liver, and testis, TEProF2 detected far fewer TE-derived TSSs than our RAMPAGE-based pipeline (Supplementary Fig. 3A). Nevertheless, 68–88% of TEProF2-identified sites were recovered by our pipeline, indicating high concordance and demonstrating that RAMPAGE substantially improves sensitivity for promoter detection. We further benchmarked our TE-derived TSSs against other TSS-mapping assays and curated promoter catalogs. A clear majority of TE-derived TSSs overlapped GRO-cap[30] and PacBio Iso-Seq[31], showing greater concordance than other TE-overlapping TSSs (Supplementary Fig. 3B). Consistent results were obtained when compared with FANTOM CAGE[32], RAMPAGE rPeaks[25], TE-TSS[16], and cCREs-PLS[33](Supplementary Figs. 3C–F). Specifically, 84% of GM12878 and 91% of K562 high-confidence TE-derived TSSs were supported by GRO-cap or Iso-Seq, and an additional 7% and 4% coincided only with existing promoter annotations (Supplementary Fig. 3G). Notably, our pipeline uncovered 46 and 33 novel high-confidence TE-derived TSSs in GM12878 and K562, respectively, all lacking any prior annotation.

Collectively, these results validate the authenticity, robustness, and biological relevance of our TE-derived promoters.

## A pan-tissue atlas of human TE-derived TSSs reveals extensive tissue specificity and functional relevance

By applying our pipeline to 115 previously curated high-quality RAMPAGE datasets[25] spanning 87 tissues and 28 cell types (Supplementary Data 1), we identified 26,056 TE-derived TSSs expressed in at least one biosample, of which 13,552 met our high-confidence threshold (Fig. 2A). Of all TE-derived TSSs, 15,222 (58%) were detected in only a single biosample, and 7575 (56%) of high-confidence sites were likewise sample-specific. On average, each biosample contained 622 total and 355 high-confidence TE-derived TSSs (Fig. 2A). Within the high-confidence subset, each sample harbored a mean of 274 annotated and 76 novel TSSs, with the novel sites displaying significantly higher expression than annotated ones (median RPM 1.93 vs. 1.00; one-sided Wilcoxon rank-sum test *P*-value < $2.2 \times 10^{-16}$; Supplementary Fig. 4A). Notably, testis samples exhibited the highest number of sample-specific TSSs, accounting for 20% of all and 26% of high-confidence sample-specific sites, underscoring testis as a hotspot for TE-initiated transcription (Fig. 2A).

To assess their genomic origins and potential functional roles, we mapped each TE-derived TSS to its host transcript. Protein-coding genes and long non-coding RNAs (lncRNAs) accounted for 65% and 15% of all TE-derived TSSs, respectively, and for 52% and 19% of high-confidence sites (Fig. 2B; Supplementary Fig. 4B, C). The remaining TSSs

mainly mapped to de novo assembled transcripts lacking existing annotation, representing 15% of total and 24% of high-confidence TSSs —a pattern most pronounced in testis. Our prior study using limited RNA-seq data showed that genes with TE-derived TSSs were both highly tissue-specific and enriched among highly expressed genes[16]. With this expanded atlas, we confirmed that genes with TE-derived TSSs are enriched among highly expressed genes (FPKM ≥ 10), comprising 5% of that group, which represents a ~1.5-fold enrichment relative to all expressed genes (Fig. 2C). Both high-confidence and all TE-derived TSS–associated genes exhibited significantly greater tissue specificity than housekeeping genes (Fig. 2D; one-sided Wilcoxon rank-sum test *P*-values < $2.2 \times 10^{-16}$).

Given these gene-level observations, we investigated whether TE-derived TSSs themselves display tissue-specific expression. Clustering a subset of 18 biosamples (nine tissues, two donors) by TE-derived TSS profiles (Supplementary Fig. 4D) revealed tighter correlations within the same tissue (median Jaccard similarity = 0.32) than between different tissues (median Jaccard similarity = 0.09; one-sided Wilcoxon rank-sum test *P*-value = $1.3 \times 10^{-6}$). Extending clustering to all 87 biosamples spanning 14 tissues again grouped biologically related tissues together (Fig. 2E; Supplementary Fig. 4E). Finally, Gene Ontology (GO) enrichment of genes harboring expressed TE-derived TSSs uncovered clear, tissue-relevant functions, including synapse part and postsynaptic density in brain, acute inflammatory response and complement activation in lung, contractile fiber and sarcomere in testis, and immune response and vesicle-mediated transport in spleen (Fig. 2E). Together, these results demonstrate that TE-derived TSSs not only expand transcript diversity but also carry strong, tissue-specific functional signals.

## Origins and characteristics of transposable elements giving rise to TE-derived TSSs

Previously, we showed that LTR elements are the most enriched source of TE-derived TSSs[16]. To dissect positional and lineage preferences, we first compared the distributions of TE-derived TSSs and annotated TEs around genes (Supplementary Fig. 5A). Consistent with prior reports that TEs are generally excluded from core promoters due to purifying selection[34], genomic TEs are significantly depleted near promoters. In contrast, TE-derived TSSs exhibit strong promoter-proximal enrichment, most prominently in testis samples, suggesting a lineage- and tissue-specific regulatory repurposing of these elements. We then quantified the TE classes contributing to TSSs and found that SINEs, LINEs, and LTRs collectively account for the majority of TE-derived TSSs, with smaller contributions from DNA transposons and SVAs (Fig. 3A). Across most tissues, seven core TE subfamilies—ERV1, Alu, L1, MIR, L2, ERVL, and ERVL-MaLR—are consistently over-represented, with testis engaging the widest variety of TE classes (Supplementary

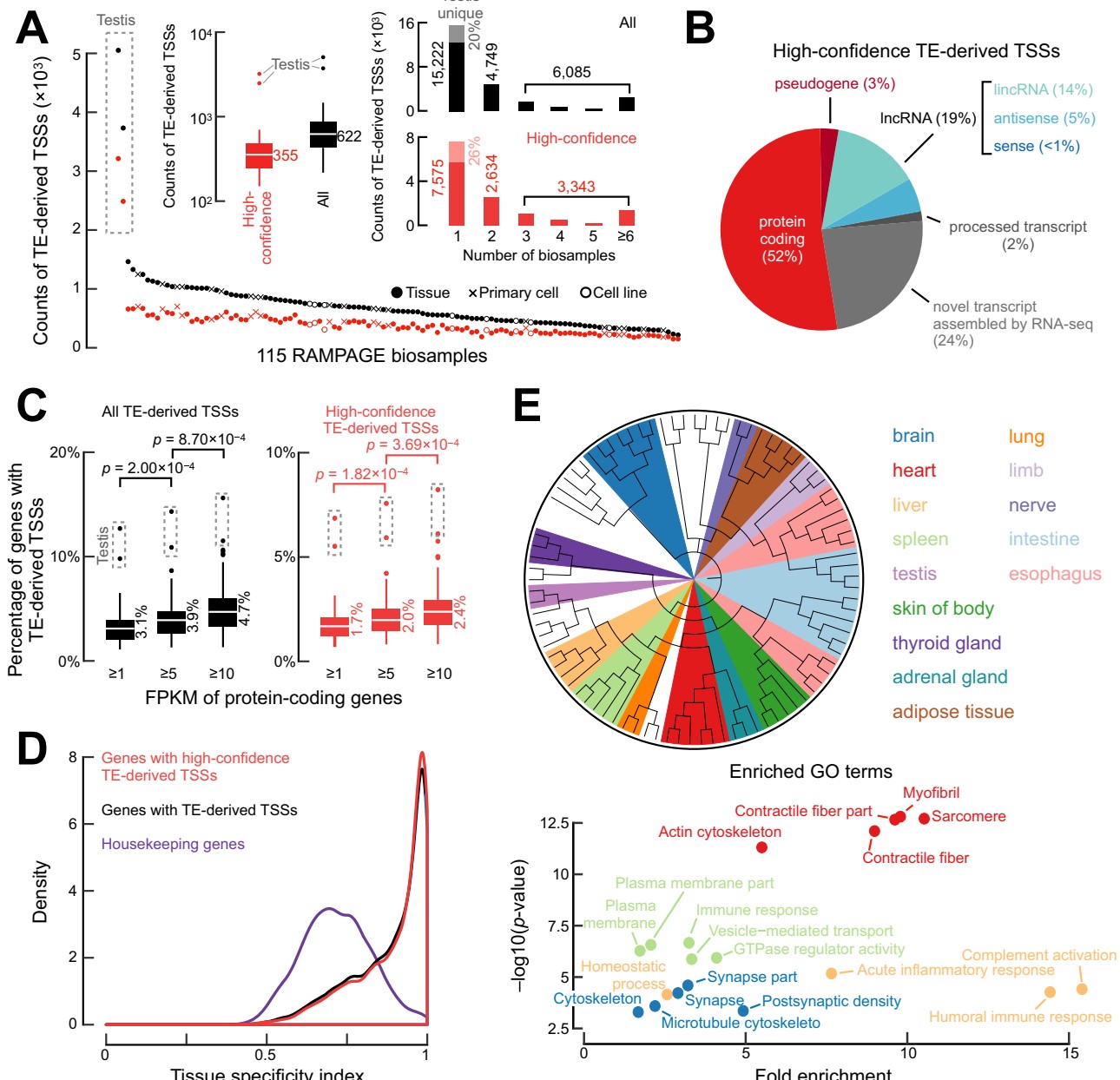

**Fig. 2 | Characteristics of TE-derived TSSs and their host genes. A** Numbers of TE-derived TSSs identified using RAMPAGE data in 115 biosamples. Each biosample is represented by one black dot (all TE-derived TSSs) and one red dot (high-confidence TE-derived TSSs). The two biosamples with the most TE-derived TSSs (encased in a dashed rectangle) were testis tissues from two donors. The top-left inset indicates that more than half of TE-derived TSSs in each biosample belong to the high-confidence subset, and the top-right inset contains two histograms showing counts of TE-derived TSSs identified in different numbers of biosamples. In each boxplot, the center line indicates the median, the box represents the interquartile range, and the whiskers extend to the most extreme values within 1.5× the interquartile range. **B** A pie chart tallies the types of genes linked to the high-confidence TE-derived TSSs. Half of the high-confidence TE-derived TSSs are connected to the transcripts of protein-coding genes. **C** The protein-coding genes with higher expression levels are more likely to have TE-derived TSSs. Each boxplot summarizes data from 115 biosamples, with the center line indicating the median, the box representing the interquartile range, and the whiskers extending to the most extreme values within 1.5× the interquartile range. One-sided Wilcoxon rank-sum test *P*-values are shown. **D** Genes with TE-derived TSSs exhibit significantly higher tissue specificity than housekeeping genes. **E** (Top) Dendrogram resulting from agglomerative hierarchical clustering of all tissue samples with RAMPAGE data based on their expression profiles of TE-derived TSSs. Each leaf of the tree represents one tissue sample, and subtrees dominated by a single tissue type are highlighted. (Bottom) Gene ontology (GO) analysis of genes with tissue-specific TE-derived TSSs detected in each tissue type. Top 5 enriched GO terms for each tissue type, with fold enrichment and *P*-values derived from Fisher's exact test, are shown.

Fig. 5B). Within this core set, Alu and MIR contribute the majority of SINE-derived TSSs; L1 and L2 dominate LINE-derived sites; and LTR-derived TSSs originate predominantly from ERV1, followed by ERVL-MaLR and ERVL, with ERVK contributing the least (Fig. 3A). Finally, comparing TE-derived TSSs to all genomic TEs revealed that only ERV1, MIR, and L2 are significantly over-represented. Conversely, despite their high copy number, Alu and L1 are under-represented relative to the genomic background, whereas all other subfamilies show near-background or modest enrichment levels (Fig. 3B).

These subfamily-level patterns prompted us to ask whether TE-derived TSSs differ in their evolutionary origins and structural properties. To determine evolutionary age, we classified each

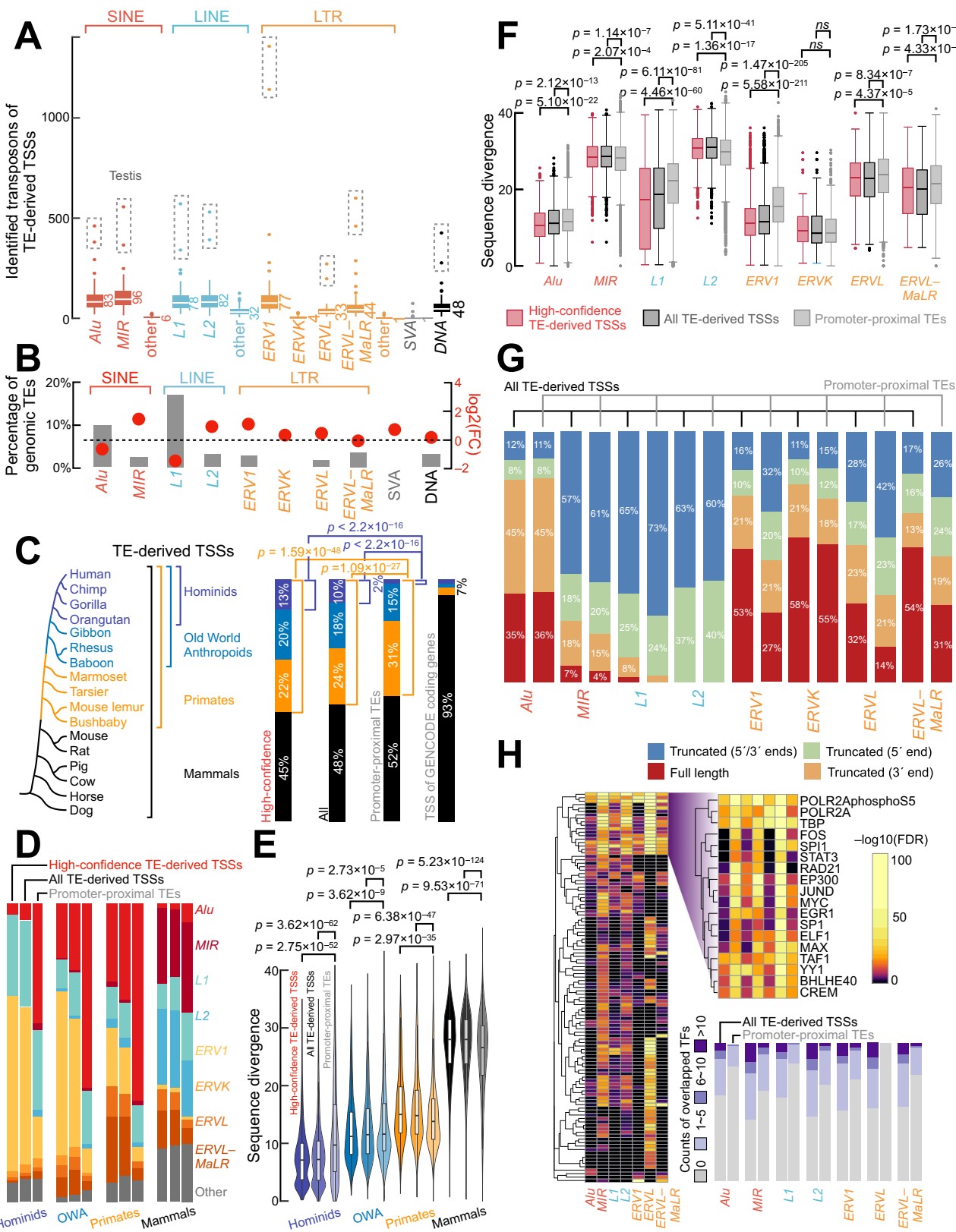

TE-derived TSS into one of four age categories based on its presence–absence patterns across different mammalian genomes[35] (Fig. 3C; see Methods). TSSs present in any non-primate mammalian genome were classified as "Mammals"; those restricted to primates as "Primates"; those shared by Old World Anthropoids as "OWA"; and those unique to the human–great ape lineage as "Hominids". Unlike canonical TSSs, 93% of which fall into the ancient "Mammals" category,

~52% of TE-derived TSSs are primate-specific (encompassing the "Primates", "OWA", or "Hominids" categories), a shift toward recent insertions that echoes TE-driven regulatory innovation in primates[10,17]. Compared with promoter-proximal TEs, both primate-specific and "hominid" categories are more frequent among TE-derived TSSs. Specifically, 55% of high-confidence and 52% of all TE-derived TSSs fall into the primate-specific category, compared with 48% of promoter-

**Fig. 3 | TE-derived TSSs arise from specific TE families with different evolutionary ages. A** A series of boxplots that highlight the distribution of TE families and subfamilies from which TE-derived TSSs arise. Each boxplot summarizes data from 115 biosamples, with the center line indicating the median, the box representing the interquartile range, and the whiskers extending to the most extreme values within 1.5× the interquartile range. The testis tissues (encased in a dashed rectangle) contained the most TE-derived TSSs in most TE families or subfamilies. **B** Fold enrichments (red dots) and genomic percentages (gray bars) of TE subfamilies from which TE-derived TSSs arise. **C** According to evolutionary age, TE-derived TSSs, promoter-proximal TEs, and TSSs of GENCODE protein-coding genes were clustered into "Hominids", "Old World Anthropoids (OWA)", "Primates", and "Mammals" categories (see Methods for more details). Chi-squared test P-values are shown. **D** Distribution of TE subfamilies in different evolutionary categories for high-confidence TE-derived TSSs, all TE-derived TSSs, and promoter-proximal TEs. **E** TE-derived TSSs showed lower sequence divergence than promoter-proximal TEs in evolutionarily young categories ("Hominids" and "OWA"), while in evolutionarily old categories ("Primates" and "Mammals"), TE-derived TSSs had higher sequence divergence than promoter-proximal TEs. In each boxplot, the center line indicates the median, the box represents the interquartile range, and the whiskers extend to the most extreme values within 1.5× the interquartile range. One-sided Wilcoxon rank-sum test P-values are shown. **F** Boxplots comparing the sequence divergence between TE-derived TSSs and promoter-proximal TEs in different TE subfamilies. In each boxplot, the center line indicates the median, the box represents the interquartile range, and the whiskers extend to the most extreme values within 1.5× the interquartile range. One-sided Wilcoxon rank-sum test P-values are shown. **G** Structural completeness of TE subfamilies for TE-derived TSSs and promoter-proximal TEs. **H** Transcription factor enrichment at TE-derived TSSs for each TE subfamily (Fisher's exact test FDR values were color-coded). Enriched transcription factors shared by different TE subfamilies were shown in the zoomed-in view. The bottom-right inset contains histograms showing counts of overlapped transcription factors in TE-derived TSSs and promoter-proximal TEs for different TE subfamilies.

proximal TEs (Chi-squared test $P$-values $< 1.59 \times 10^{-48}$). Furthermore, 13% of high-confidence and 10% of all TE-derived TSSs belong to the "Hominids" category, a substantial increase compared with only 2% of promoter-proximal TEs (Chi-squared test $P$-values $< 2.2 \times 10^{-16}$), suggesting recent TE insertions may contribute to lineage-specific regulatory innovation. This primate bias is equally pronounced in testis-specific TE-derived TSSs (Supplementary Fig. 5C).

We then examined the composition of TE subfamilies giving rise to these TSSs (Fig. 3D). Consistent with our age classification, ERV1 elements predominate in the "Hominids" and "OWA" categories, whereas Alu elements are most frequent in the "OWA" and "Primates" categories. In contrast, the ancient families MIR and L2 are largely confined to the "Mammals" category. When compared with promoter-proximal TEs, ERV1 is significantly enriched and Alu significantly depleted in primate-specific lineages, while ERVL and ERVL-MaLR elements show enrichment in non-hominid groups. Notably, the ERV1 enrichment and Alu depletion patterns are further amplified in testis-specific TE-derived TSSs (Supplementary Fig. 5C). To complement the phylogenetic classification, we examined sequence divergence as an orthogonal proxy for evolutionary age, with lower divergence indicating more recent origin. Divergence among TE-derived TSSs is lowest in the "Hominids" category and increases through "OWA", "Primates", to "Mammals" categories (Fig. 3E), consistent with younger elements being co-opted more recently[36,37]. When compared with promoter-proximal TEs, TE-derived TSSs in the "Hominids" and "OWA" categories show reduced sequence divergence, whereas those in the "Primates" and "Mammals" categories display higher divergence. At the subfamily level, Alu, L1, ERV1, ERVL, and ERVL-MaLR elements at TE-derived TSSs are less divergent than their promoter-proximal counterparts, while MIR and L2 elements are more divergent (Fig. 3F). Notably, in testis-specific TE-derived TSSs, this reduction in divergence is significant only for Alu and ERV1 elements (Supplementary Fig. 5D).

Because evolutionary age may influence not only sequence divergence but also copy integrity, we examined the structural completeness of TE copies contributing to TE-derived TSSs (Fig. 3G). In the non-LTR TE families MIR, L1, and L2, 93%, 98%, and 100% of TE-derived TSSs, respectively, derive from truncated copies. By contrast, in the LTR families ERV1, ERVL, and ERVL-MaLR, 53%, 32%, and 54% of TE-derived TSSs originate from full-length insertions. For comparison, only 27% of promoter-proximal ERV1, 14% of ERVL, and 31% of ERVL-MaLR elements are full-length. These data suggest that, although truncated non-LTR fragments can initiate transcription, full-length LTRs are preferentially retained or co-opted as promoters.

To assess whether lineage- and subfamily-specific TE-derived TSSs correspond to functional divergence in host genes, we performed GO enrichment analysis. Genes associated with hominid-specific TE-derived TSSs were enriched for neural and immune programs and for cardiac or muscle-related development (Supplementary Fig. 5E).

Primate-specific TE-derived TSSs showed enrichment for immune, developmental, and metabolic processes (Supplementary Fig. 5E). Because ERV1 elements are strongly over-represented among primate-specific TSSs, we further examined genes associated with primate-specific ERV1-derived TSSs and found enrichment for interferon signaling, cell–cell adhesion, long-chain fatty acid metabolism, xenobiotic response, and transposable element silencing (Supplementary Fig. 5F). Together, these results indicate that both lineage origin and TE subfamily composition shape the functional diversification of TE-derived promoters, with primate ERV1 insertions particularly contributing to immune and metabolic regulatory innovation.

Finally, to explore which transcription factors (TFs) are preferentially associated with TE-derived TSSs, we overlapped them with ChIP-seq peaks from 694 TFs curated by the ENCODE Consortium[38] (Supplementary Fig. 6A). TE-derived TSSs tend to be bound by a broader repertoire of TFs compared to promoter-proximal TEs (Chi-squared test, $p < 2.2 \times 10^{-16}$; Fig. 3H). Several general TFs—including POLR2A, TBP, TAFs, and YY1—show enrichment across TE-derived TSSs from diverse subfamilies (Fig. 3H). Aggregate ChIP-seq signal profiles in the matched biosample further support stronger binding of these TFs at TE-derived TSSs relative to promoter-proximal TEs (Supplementary Fig. 6B). In contrast, several TFs exhibited subfamily-specific patterns suggestive of preferential TF association, supported by both ChIP-seq signal and motif enrichment (Supplementary Fig. 6C). For instance, ETS1 and ETV5, which recognize canonical ETS motifs[39], were preferentially enriched at MIR-derived TSSs. Similarly, NFYA and NFYB, which bind CCAAT boxes[40], showed selective enrichment at ERV1- and ERVL-MaLR-derived TSSs. USF1 and USF2, which recognize E-box motifs[41], were enriched at ERVL- and ERVL-MaLR-derived TSSs. Together, these results suggest that specific TE subfamilies may retain recognizable TF motifs that could facilitate subfamily-specific regulatory activity. Notably, TF binding patterns at TE-derived TSSs appear distinct from those governing autonomous TE transcription. For example, we previously showed that AP-1 family members, such as FOS and JUN, preferentially bind Pol III-transcribed Alu elements to promote their expression[24]. However, this enrichment was absent from Alu-derived TSSs (Supplementary Fig. 6D), which is suggestive of a transition from internal TE-driven transcription to host-mediated co-option of TE fragments as regulatory modules.

## TE-derived TSSs are enriched for narrow-peak promoter architectures and contribute to gene expression

RAMPAGE enables single-nucleotide resolution mapping of transcription start sites for both autonomous TE transcripts[24] and protein-coding genes[25], making it ideally suited for quantifying promoter usage across all TSSs of a gene. To confirm the accuracy of our TE-derived TSS annotations, we overlapped our annotated TE-derived TSSs with RAMPAGE peaks from K562 and GM12878 and found that

nearly all peaks fell within ±20 nt of their corresponding TE-derived TSSs (Supplementary Fig. 7A; permutation test $P$-values $< 1 \times 10^{-4}$). Consistently, RAMPAGE peak summits at TE-derived TSSs show variability comparable to that of mRNA TSSs and are significantly less variable than those at other TE-overlapping TSSs (Supplementary Fig. 7B). We next asked whether these sites conform to established Pol II promoter architectures. Human Pol II promoters are known to adopt either a narrow shape, characterized by a focused TSS, a TATA box ~30 nt upstream, and low CpG density, typically associated with tissue-specific expression, or a broad shape, in which initiation is dispersed, CpG density is high, and TATA boxes are absent, generally driving ubiquitous transcription[42]. We therefore classified all TSSs by RAMPAGE peak width into narrow (≤10 bp) and broad (>10 bp) categories (Fig. 4A). In comparison to canonical TSSs, TE-derived TSSs are more than twice as likely to adopt a narrow peak architecture (32% versus 15%; one-sided Wilcoxon rank-sum test $P$-values $< 2.2 \times 10^{-16}$; Fig. 4A), and these narrow peaks exhibit significantly less width variability across biosamples than broad peaks (Supplementary Fig. 7C), indicating that their promoter shape is consistently maintained under diverse conditions.

To determine whether these narrow-peak sites also bear the sequence hallmarks of focused Pol II promoters, we assessed TATA box presence and CpG context. TE-derived TSSs are 1.5-fold more likely than canonical TSSs to contain a TATA box (Chi-squared test $P$-values $< 2.30 \times 10^{-48}$; Fig. 4B), with narrow-peak TE-derived TSSs harboring the motif at −25 to −30nt (Supplementary Fig. 7D), just as in typical Pol II promoters. By contrast, Pol III-transcribed Alu elements position their TATA boxes at −15 nt (Supplementary Fig. 7D), reflecting differences in the underlying transcriptional machinery. Additionally, over 93% of TE-derived TSSs lack upstream CpG islands compared with 60% of expressed canonical TSSs (Chi-squared test $P$-values $< 2.2 \times 10^{-16}$; Fig. 4B), and TE-derived TSSs display markedly lower GC content than their canonical counterparts (one-sided Wilcoxon rank-sum test $P$-values $< 2.2 \times 10^{-16}$; Supplementary Fig. 7E). Importantly, TE-derived TSSs that contain a TATA box almost always display narrow peaks, whereas those with upstream CpG islands consistently fall into the broad-peak category (Fig. 4B). Together, these observations demonstrate that, relative to canonical TSSs, TE-derived TSSs are enriched for narrow peaks and adopt the TATA-enriched, CpG-poor architecture characteristic of tissue-specific Pol II promoters.

We then quantified TE-derived TSS counts per gene and found that ~85.6% of genes possess a single TE-derived TSS, 1.1% harbor two, and just 0.3% carry three or more, with testis samples exhibiting elevated per-gene counts (Fig. 4C; Supplementary Fig. 8A). To assess how much these sites actually contribute to gene output, we defined promoter usage as the fraction of RAMPAGE tags at each TE-derived TSS divided by the total tags across all TSSs of that gene. More than 25% of genes with TE-derived TSSs obtain at least half of their promoter activity from these sites; when lowering the usage threshold to 10%, roughly half of such genes meet the criterion, indicating a substantial contribution of TE-derived TSSs to overall gene expression (Fig. 4D). To dissect their functional importance, we categorized TE-derived TSSs by usage as unique (the sole TSS of the gene), predominant (the highest usage among all TSSs), or non-predominant (all others). Although non-predominant sites constitute the majority, roughly 25% are unique and 11% predominant (Fig. 4E; Supplementary Fig. 8B). Predominant TSSs typically account for at least 50% of promoter usage, and even non-predominant sites frequently contribute more than 5% (Supplementary Fig. 8C). In support of their functional impact, unique and predominant TSSs yield significantly stronger SuRE signals than non-predominant sites (Fig. 4F).

Finally, we examined the evolutionary origin and TE family bias behind these classes. Unique and predominant TE-derived TSSs are significantly enriched for primate-specific and hominid insertions (Fig. 4G), both at the TSS (Chi-squared test $P$-values $< 2.2 \times 10^{-16}$) and host gene levels (Chi-squared test $P$-values $< 2.2 \times 10^{-16}$), suggesting that recent TE integrations have driven lineage-specific regulatory innovation. Subfamily analysis further reveals that these sites most often arise from L1 and ERV elements within primate lineages and are depleted of the older MIR and L2 families typical of mammalian genomes (Supplementary Fig. 8D). Correspondingly, they exhibit lower sequence divergence in Alu, L1, and select ERV subfamilies compared with non-predominant sites (Supplementary Fig. 8E), consistent with their more recent origin and retention of intact promoter features[43].

## Young TE-derived TSSs preserve intrinsic promoter architecture to drive precise transcription initiation

Having shown that TE-derived TSSs differ in both transcriptional activity and evolutionary divergence, we next asked whether these differences stem from preserved, intrinsic TE promoter elements or from post-integration, host-driven regulation. To estimate evolutionary age, we used the percent divergence of each TE copy from its consensus sequence as a proxy for time since insertion. If intrinsic promoter strength were preserved, it should manifest as higher SuRE activity. Indeed, SuRE signals increase stepwise from the oldest to the youngest TE lineages, indicating that lower sequence divergence is linked to stronger intrinsic promoter activity (Supplementary Fig. 9A). Highly diverged MIR and L2 families exhibit weak SuRE activity, whereas low-divergence ERV1 elements show the strongest signals (Fig. 5A; Supplementary Fig. 9B).

To directly assess whether transcription initiation occurs from conserved positions within TE sequences, we aligned each TE-derived TSS to the consensus sequence of its subfamily and calculated the standard deviation of its relative position across genomic copies. Within the Alu, L1, and ERVL–MaLR families, subfamilies with lower sequence divergence exhibited tightly clustered TSS positions, while those with higher divergence showed broader, more dispersed TSS distributions (Fig. 5A).

Subdivision of the Alu family illustrates this gradual pattern clearly (Fig. 5B). AluY, the youngest subfamily specific to hominids[44], displayed highly focal TSS positioning and the strongest SuRE signals. In contrast, the more ancient AluS and AluJ subfamilies exhibited progressively more dispersed TSS positioning and reduced SuRE activity. A similar divergence-associated hierarchy was observed in the L1 family (Fig. 5B). The human-specific L1HS subgroup retained narrow, consistent TSS placement and high SuRE activity, whereas the older L1P (primate-specific) and L1M (mammalian-wide) subfamilies showed increasingly dispersed TSSs and attenuated SuRE signals. The ERVL–MaLR superfamily followed the same evolutionary trajectory (Supplementary Fig. 9C). THE1, largely confined to primates[45], exhibited focused TSS placement and robust SuRE reporter activity, whereas the older MST and MLT1 subfamilies, present across mammals, showed broader TSS distributions and weaker SuRE output.

Together, these findings indicate that promoter precision and activity form a continuous evolutionary gradient that parallels TE sequence divergence. Elements with low divergence preserve their intrinsic promoter architecture and drive strong, well-defined initiation, while those that have accumulated substitutions and deletions exhibit dispersed transcription initiation and increasing dependence on host regulatory environments.

## Discussion

Recent advances in high-throughput sequencing have revealed that TEs play fundamental roles in the evolution of gene regulation across multicellular eukaryotes[2,43,46]. By introducing intrinsic cis-regulatory modules into host genomes, TEs have been exapted into existing gene networks, thereby increasing the regulatory dynamism of the genome[1,3]. Certain TE families have in turn evolved novel TSSs that

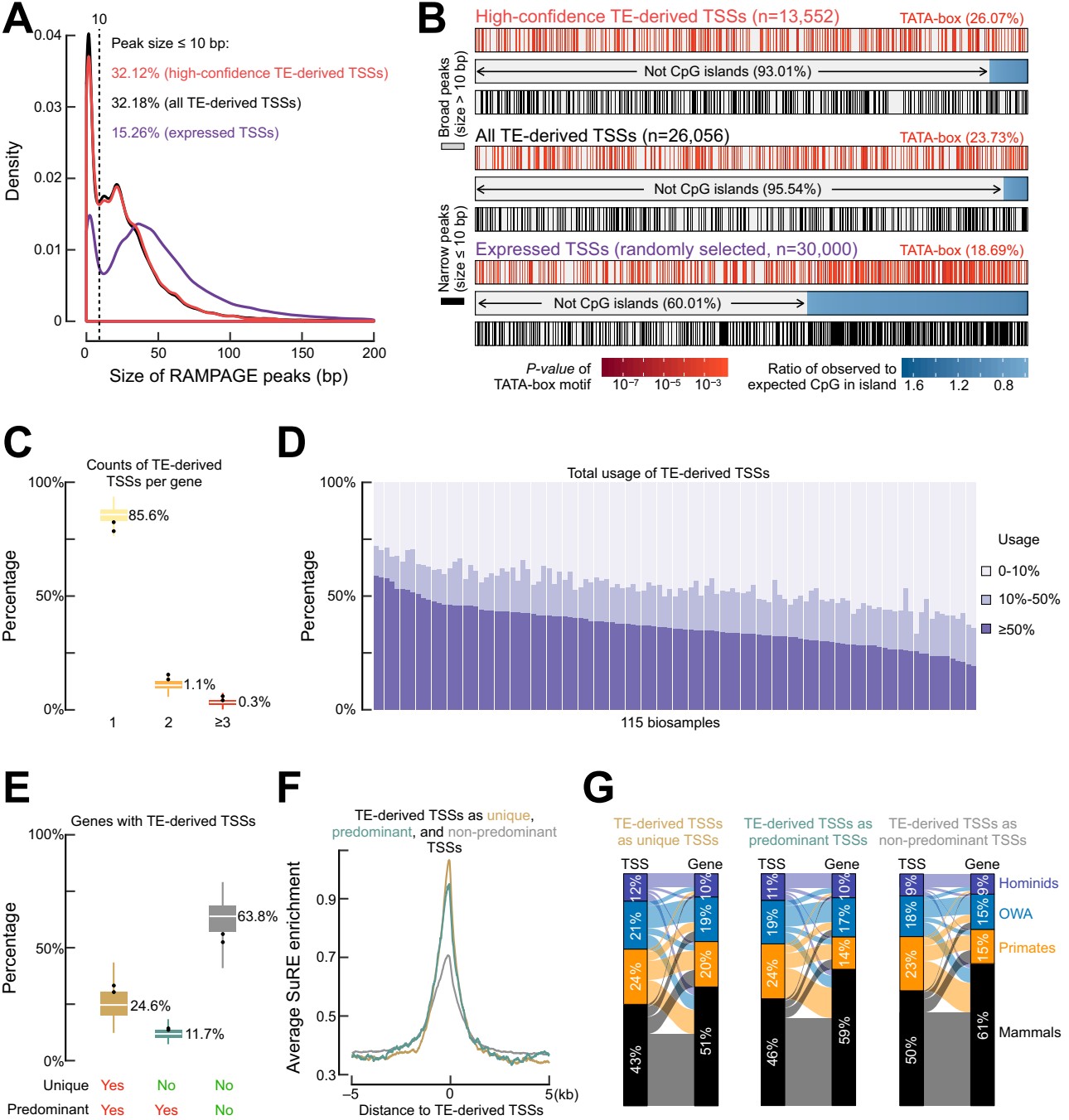

**Fig. 4 | TE-derived TSSs are enriched for narrow-peak promoter architectures and contribute substantially to gene expression. A** The size distribution of RAMPAGE peaks for high-confidence TE-derived TSSs (red line), all TE-derived TSSs (black line), and expressed TSSs (purple line). RAMPAGE peaks with size ≤10 bp were defined as narrow peaks, otherwise broad peaks. **B** Promoter types classified based on TATA-box motifs (top bar plots; motif *P*-values were calculated using FIMO), CpG islands (middle bar plots), and TSS peak type (bottom bar plots) for high-confidence TE-derived TSSs (top panel), all TE-derived TSSs (middle panel), and expressed TSSs (bottom panel). **C** Percentage of genes stratified by the number of TE-derived TSSs. Each boxplot summarizes data from 115 biosamples, with the center line indicating the median, the box representing the interquartile range, and the whiskers extending to the most extreme values within 1.5× the interquartile range. Testis samples are highlighted with black points. **D** Total usage of TE-derived TSSs per gene across 115 biosamples. **E** Percentage of genes with TE-derived TSSs classified as unique (yellow), predominant (blue), or non-predominant (gray). Each boxplot summarizes data from 115 biosamples, with the center line indicating the median, the box representing the interquartile range, and the whiskers extending to the most extreme values within 1.5× the interquartile range. Testis samples are highlighted with black points. **F** Average SuRE enrichment with respect to TE-derived TSSs classified as unique (yellow), predominant (blue), or non-predominant (gray) in K562 cells. **G** Alluvial plot showing evolutionary categories of TE-derived TSSs and their host genes by TE-derived TSS usage pattern, with TE-derived TSSs classified as unique (yellow), predominant (blue), and non-predominant (gray).

initiate alternative isoforms of conserved protein-coding genes and drive a substantial fraction of long noncoding RNAs[16,47–49].

Here, we leverage high-quality RAMPAGE data to map over 26,000 TE-derived TSSs at single-nucleotide resolution across 115 human biosamples spanning 87 tissues and 28 cell types, unambiguously linking each initiation event to its parent transcript. This precision is essential for resolving TE-derived promoters within highly repetitive regions and for accurately quantifying their usage.

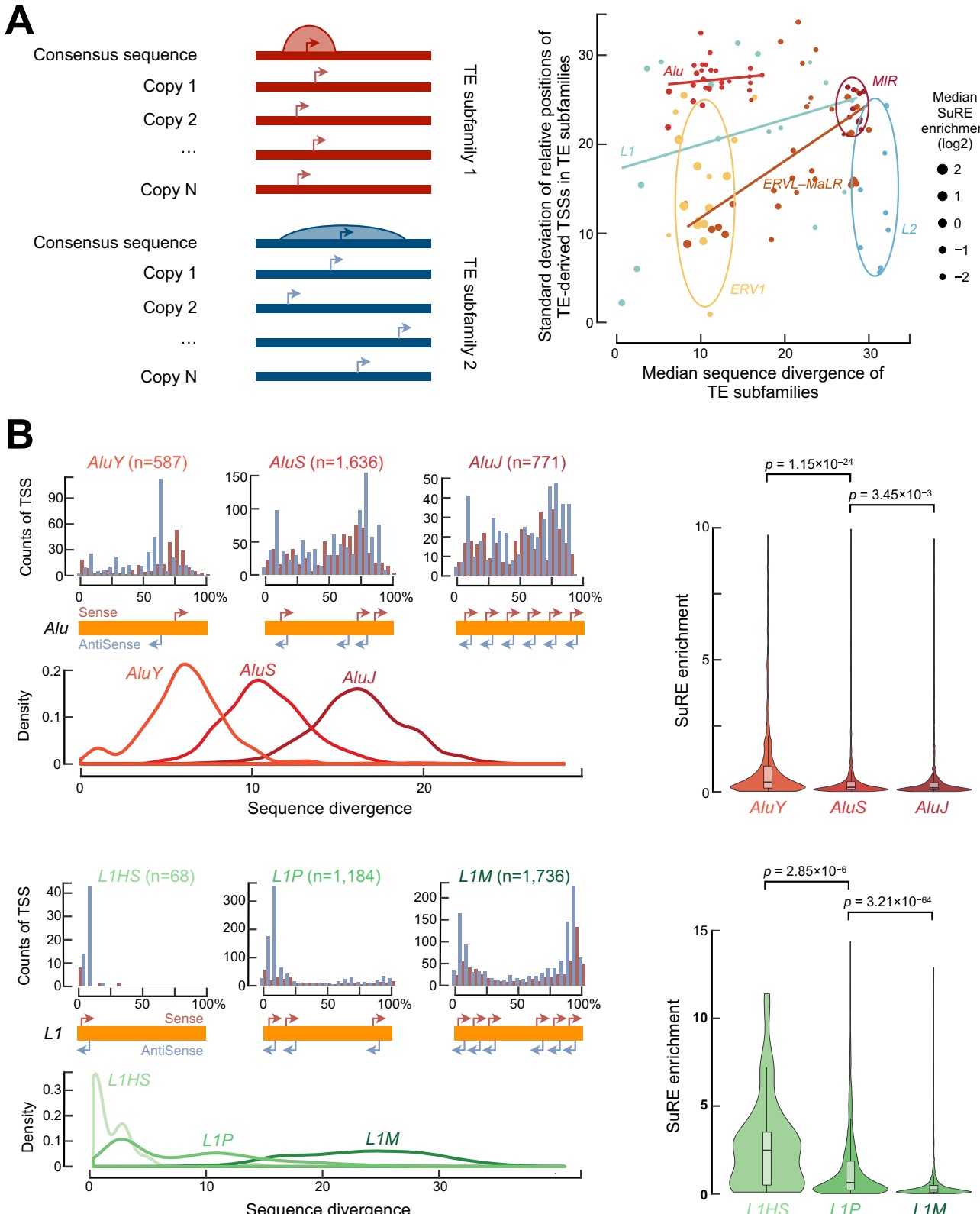

**Fig. 5 | Young TE-derived TSSs preserve intrinsic promoter architecture to drive precise transcription initiation. A** Different subfamilies of TEs showed distinct distributions of TSS positional patterns (y-axis, standard deviation of relative TSS positions) and TSS potential (dot size, median SuRE enrichment) with respective to evolutionary age (x-axis, median sequence divergence). Each dot represents one TE subfamily, with statistics calculated across all TE-derived TSSs within that subfamily. **B** TSS positions (left top), sequence divergence (left bottom),

and SuRE enrichment profiles (right) of TE-derived TSSs arising from different Alu (AluY, $n = 587$; AluS, $n = 1636$; AluJ, $n = 771$; top panel) and L1 (L1H, $n = 68$; L1P, $n = 1184$; L1M, $n = 1736$; bottom panel) subfamilies. In each boxplot, the center line indicates the median, the box represents the interquartile range, and the whiskers extend to the most extreme values within 1.5× the interquartile range. One-sided Wilcoxon rank-sum test $P$-values are shown.

Our pan-tissue atlas reveals that TE-derived TSSs act as dynamic regulators of tissue-specific gene expression rather than passive genomic vestiges. On average, each biosample contains 622 TE-derived TSSs. Although TE-derived TSSs appear in only ~5% of highly expressed genes, their contribution at the level of individual genes can still be substantial. Approximately one quarter of genes with TE-derived TSSs derive more than half of their promoter activity from these sites, and roughly half show at least 10% usage, with over 30% of TE-derived TSSs functioning as the predominant or unique promoter. These findings indicate that, despite their limited genome-wide prevalence, TE-derived promoters often act as dominant regulatory elements for a considerable subset of genes, thereby contributing to transcriptomic diversity and regulatory innovation in specific genomic contexts. These findings confirm and extend earlier reports of TE domestication as alternative promoters[16,47–49] and underscore their substantial impact on transcriptomic complexity. Compared with canonical TSSs, TE-derived TSSs are significantly enriched in narrow-peak architectures marked by TATA-box motifs, low CpG density, and focused initiation—hallmarks of tissue-specific RNA polymerase II promoters[42,50]. This specialized architecture enables precise spatial and temporal regulation, as reflected both in the intrinsic activity of TE-derived TSSs and in the highly restricted expression profiles of their host genes. Clustering based on TE-derived TSS usage groups biologically related samples together, and functional enrichment of host genes highlights clear, tissue-relevant functions.

Notably, testis emerges as a pronounced hotspot for TE-derived initiation. Testicular tissue harbors the greatest abundance and diversity of TE-derived TSSs, which frequently serve as the unique or predominant initiation site for host genes. This enrichment likely reflects the permissive chromatin state and specialized transcription factor milieu of male germ cells, which facilitate the co-option of TE promoters[51,52]. Moreover, testis-specific TSSs often originate from younger TE subfamilies that retain stronger intrinsic promoter activity, driving germline-restricted transcriptional diversity and regulatory innovation.

Our evolutionary analysis reveals a continuous gradient in the transcriptional precision and strength of TE-derived TSSs that parallels sequence divergence and phylogenetic age. Younger TE subfamilies restricted to primates and hominids retain intact core promoter motifs and drive tightly clustered initiation at conserved positions, resulting in precise and robust transcription. In contrast, older lineages such as MIR and L2 exhibit higher sequence divergence, more dispersed TSS placement, weaker intrinsic strength, and partial loss of structural integrity, reflecting gradual promoter decay through sequence erosion and selection[18,35]. Previous reports have demonstrated that primate-specific TEs often harbor lineage-restricted transcription factor binding sites active during early development[53,54], consistent with the longer retention of functional motifs in younger elements. The strong correlation between TSS clustering and promoter strength across Alu, L1, and ERVL–MaLR families further supports a model in which promoter exaptation follows a continuous evolutionary trajectory, shaped by the preservation of intrinsic sequence features in younger TEs and by the surrounding genomic context in older insertions.

To explore the functional consequences of this evolutionary diversification, we examined lineage- and subfamily-specific TE-derived TSSs. Our analyses suggest that hominid- and primate-specific TE insertions have been selectively co-opted into regulatory programs controlling neural development, immune responses, and metabolic processes. In particular, primate-specific ERV1 elements appear to contribute disproportionately to interferon signaling, lipid metabolism, and TE silencing pathways, which may have facilitated species-specific innovations in immune defense and metabolic adaptation. This selective integration indicates that TE exaptation is not uniform across the genome; rather, the lineage origin and subfamily identity of each insertion influence the host pathways it regulates. Such patterns

likely reflect both the intrinsic promoter features retained by younger TEs and their compatibility with pre-existing transcriptional networks, ultimately enabling TE-derived promoters to drive regulatory innovations that could underpin primate-specific traits.

Together, our findings deepen the understanding of how mobile elements are co-opted as functional promoters to diversify transcriptional regulation. By mapping TE-derived TSSs at single-nucleotide resolution and quantifying their usage across an unprecedented range of human tissues, we reveal their capacity to act as both evolutionary innovators and tissue-specific regulators within complex gene networks. This comprehensive atlas will serve as a valuable resource for future studies of promoter evolution and gene regulatory innovation in primates and beyond.

## Methods

### Computational pipeline to annotate TE-derived TSSs using RAMPAGE data

RAMPAGE peaks were first called and filtered (left panel in Supplementary Fig. 1A) as previously described[24,25]. Briefly, we excluded RAMPAGE datasets with non-redundancy fraction <0.25 and retained 115 RAMPAGE datasets as high-quality datasets (Supplementary Data 1). BAM files for these datasets, aligned to the GRCh38/hg38 reference human genome using STAR[55] based on the ENCODE RAMPAGE Processing Pipeline (https://www.encodeproject.org/data-standards/rampage), were downloaded from the ENCODE Portal (https://www.encodeproject.org). Properly aligned read pairs (R1 and R2 denote each mate of the read pair) with uniquely aligned R2 reads were collapsed with the same alignment coordinates and the identical 15-bp barcode at the 5′-end of R2 reads to remove PCR duplicates. Read pairs from biological replicates were pooled together after the PCR duplicate removal. RAMPAGE peaks were clustered with the 5′-most base of aligned R1 reads using F-seq[56] (parameter: feature length = 30 and fragment size = 0), and resized to the length that included 70% of the reads in each original peak to eliminate lengthy tails. RAMPAGE peaks were then filtered with entropy (E) ≥ 2.5 based on the coordinates of R2 reads in each peak[24]. Peaks overlapping transposon elements (hg38 rmsk.txt downloaded from UCSC) were defined as TE-overlapping TSSs.

With downstream R2 reads, RAMPAGE peaks were assigned to annotated genes from GENCODE (V29) or RefSeq (Curated, updated on 07 April 2018) and de novo assembled transcripts (StringTie[57] with default parameters) from matched RNA-seq data (Supplementary Data 1). Candidate TE-derived TSSs (right panel in Supplementary Fig. 1A; Supplementary Data 2) were selected if (i) paired RAMPAGE reads linked the peak to exonic regions of annotated or de novo transcripts, (ii) the genomic span of the read pair was >1 kb, and (iii) the exon-assignment fraction ≥0.5 (Supplementary Fig. 1B, C). Sites located within 1 kb of annotated GENCODE/RefSeq TSSs or de novo RNA-seq–assembled TSSs were classified as high-confidence TE-derived TSSs; remaining sites were labeled RAMPAGE-only (Supplementary Fig. 1D).

### Comparison of TE-derived TSS with other TSS annotations and RNA-seq–based detection method

TSS annotations defined using GRO-cap peaks[30] (Supplementary Fig. 3B), PacBio Iso-Seq[31] (Supplementary Fig. 3B), FANTOM CAGE[32] (Supplementary Fig. 3C), RAMPAGE rPeaks[16,25] (Supplementary Fig. 3D), TE-TSS database[16] (Supplementary Fig. 3E), and cCREs-PLS V4[33] (Supplementary Fig. 3F) were downloaded and processed, respectively, and overlapped with TE-derived TSSs in corresponding cell types. To be specific, GRO-cap peaks in GM12878 and K562 cells (GEO: GSE60456) were aligned to GRCh38/hg38 reference human genome using STAR with default parameters and called using F-seq (parameter: feature length = 30 and fragment size = 0). Assembled isoforms of PacBio ISO-Seq in GM12878 and K562 cells processed using

TALON[58] were downloaded from the ENCODE Portal (https://www.encodeproject.org), and the 5′-end of assembled isoforms were regarded as the TSSs. CAGE peaks called for GRCh38/hg38 reference human genome by FANTOM Consortium were downloaded from FANTOM5 data website (https://fantom.gsc.riken.jp/5/datafiles/reprocessed). RAMPAGE rPeaks were processed according to the rPeak pipeline using 115 RAMPAGE biosamples. TE-derived TSS annotations were downloaded from the TE-TSS database (http://xozhanglab.com/TETSS). The ENCODE cell type-agnostic candidate Cis-Regulatory Elements with promoter-like signatures (cCREs-PLS) were downloaded from the SCREEN resource (http://screen.encodeproject.org). In addition, to identify TE-derived transcripts directly from RNA-seq data, we employed TEProF2 using matched RNA-seq datasets. A TE-derived transcript was considered present if it showed an expression level ≥1 TPM and at least one unique read spanning the junction between the TE and the host transcript, as previously described[29].

## Mapping and normalization of SuRE data
Raw SuRE reads (iPCR, cDNA and plasmid reads) were downloaded from GSE78709 and processed as previously reported[26], but re-mapped to the GRCh38/hg38 reference human genome. Reads of iPCR are combinations of unique barcode sequence and respective genomic DNA sequence for each SuRE fragment, and reads of cDNA and plasmid only record counts of barcode sequences in the SuRE library and the "input" library, respectively. Paired-end reads of iPCR and single-end reads of cDNA and plasmid were processed to remove adapter and vector backbone sequences using cutadapt. To associate barcode sequences with genomic coordinates, trimmed read pairs of iPCR were aligned to the GRCh38/hg38 reference human genome using Bowtie2[59] permitting a maximum insert length of 4 kb. For each SuRE fragment, cDNA and plasmid read counts from biological replicates were aggregated, and cDNA counts were normalized by plasmid counts ("input") to obtain the SuRE enrichment profiles.

## Tissue specificity analysis
We evaluated the tissue specificity of genes harboring TE-derived TSSs using a previously defined tissue specificity score[60], calculated as:

$$Tissue\ specificity\ score = \frac{\sum_{i=1}^{N} 1 - x_i}{N - 1} \tag{1}$$

where $N$ denotes the total number of biosamples and $x_i$ represents the expression level in biosample $i$, normalized by the maximal expression value across all biosamples. This tissue specificity index ranges from 0 to 1, with higher values indicating greater tissue specificity.

## Enrichment analysis of TE subfamilies and transcription factors for TE-derived TSSs
To check which TE subfamilies contribute to TE-derived TSSs, we calculated the fold enrichment for each TE subfamily in the formation of TE-derived TSSs over the genomic background. Because TE-derived TSSs are specifically enriched near TSSs (Supplementary Fig. 5A), fold enrichments were also normalized by promoter-proximal TEs (TEs in TSS ± 1 kb). To calculate FDRs, P-values from hypergeometric test and binomial test were combined by Stouffer's Z-score method, and combined P-values were corrected using the Benjamini–Hochberg procedure.

ChIP-seq peaks of TFs from the ENCODE Consortium were downloaded from Factorbook[38] (https://www.factorbook.org). Significant ChIP-seq peaks (FDR < 0.05) overlapping with TEs of TE-derived TSSs were counted, and promoter-proximal TEs (TEs in TSS ± 1 kb) were collected as the genomic control. Enrichment P-values were calculated using Fisher's exact test, and FDR correction with the Benjamini–Hochberg procedure was used to select enriched transcription factors.

To evaluate enrichment of specific transcription factor motifs at TE-derived TSSs, we applied the AME algorithm[61] from the MEME Suite to scan curated motifs from the Factorbook database[38]. Motif occurrences were quantified in both TE-derived TSS sequences and promoter-proximal TE sequences, and the enrichment was determined by comparing their frequencies between these two sets.

## Evolutionary analysis of TE-derived TSSs and host genes
To categorize TE-derived TSSs by evolutionary age, TE-derived TSSs were aligned to 16 other genomes using the liftOver tool with whole-genome alignment-chain files downloaded from the UCSC. To reduce potential false positives resulting from alignments of paralogous loci between two genomes, a minimum liftOver mapping ratio of 90% (-minMatch = 0.9) was required for TE-derived TSSs (TSS ± 10 bp) and a minimum liftOver mapping ratio of 50% (-minMatch = 0.5) was needed for TE-derived TSS surrounding regions (TSS ± 100 bp), and we also required a minimum alignment chain size of 10 kb for both target and query genomes (-minChainT = 10000 -minChainQ = 10000 -multiple). TE-derived TSSs that could not be lifted over to another genome were considered missing in that genome. Evolutionary categories and the mammalian phylogeny were defined as previously reported[35]. According to the presence or absence patterns in specific genomes, TE-derived TSSs were classified into four categories of different evolutionary ages (Fig. 3C; Supplementary Data 3): 1. TSSs whose sequences could be lifted over to at least one non-primate genome were categorized into the "Mammals" category; 2. TSSs whose sequences occurred during early primate evolution but before the last common ancestor of Old World Anthropoids (OWA) were categorized into the "Primates" category; 3. TSSs whose sequences occurred during the evolution of OWA but before the last common ancestor of hominids were categorized into the "OWA" category; 4. TSSs whose sequences occurred since the emergence of hominids were categorized into the "Hominids" category.

Because TE-derived TSSs were enriched close to known TSSs (Supplementary Fig. 5A), we selected GENCODE TSSs and nearby TEs (TSS ± 1 kb, promoter-proximal TEs) for comparison. For GENCODE TSSs, the same criteria as TE-derived TSSs were used. For nearby TEs, we required a minimum liftOver mapping ratio of 50% (-minMatch = 0.5) and a minimum alignment chain size of 10 kb for both target and query genomes (-minChainT = 10000 -minChainQ = 10000 -multiple). We used the same method as TE-derived TSSs to cluster GENCODE TSSs and promoter-proximal TEs into corresponding evolutionary categories.

To estimate the evolutionary age of host genes, we applied the same liftOver-based alignment strategy across the 16 mammalian genomes as described above. Assembled gene sequences of TE-derived TSSs were lifted over to each target genome using liftOver with a minimum nucleotide mapping ratio of 80% (−minMatch = 0.8), a minimum exon alignment ratio of 50% (−minBlocks = 0.5), and allowance for multiple mappings (−multiple). Genes failing to meet these criteria in a given genome were considered absent. Based on their presence−absence patterns, each host gene was then assigned to one of four evolutionary age categories (mammals, primates, OWA or hominid).

## Characterization of promoter features for TE-derived TSSs
TE-derived TSSs were classified into narrow peaks (size ≤ 10 bp) and broad peaks (size > 10 bp) according to the sizes of RAMPAGE peaks (Fig. 4A). To test whether a TE-derived TSS contained a TATA-box, we scanned the binding motif of TBP (MA0108.2 in JASPAR database[62]) within 25−35 bp upstream of the summit of TE-derived TSS RAMPAGE peaks using FIMO[63] (parameter: --thresh 1e-2). TE-derived TSSs were overlapped with CpG island annotations

(hg38 cpgIslandExtUnmasked.txt downloaded from UCSC) to determine whether one TE-derived TSS is located within a CpG island.

## Luciferase reporter assay

Luciferase reporter assay was performed as previously described with some modifications[27]. Sequences of TE-derived TSSs and their antisense counterparts were cloned into pGL3-basic luciferase reporter plasmids. 293FT cells were co-transfected with the luciferase reporter plasmid, a FLAG expression plasmid, and the pRL-TK Renilla luciferase plasmid using Lipo8000 (Beyotime, C0533). After 24 h of transfection, cells were lysed and luciferase activity was measured using the Dual Luciferase Reporter Assay Kit (Vazyme, DL101). Relative luciferase signal was normalized to Renilla luciferase activity for each sample. Primers used for reporter vector construction are listed in Supplementary Data 4.

## Cell culture and experimental validation of transcripts with TE-derived TSSs

K562 cells were cultured at 37 °C with 5% $CO_2$ in complete medium. Total RNA was isolated using TRIzol. TE-derived TSSs were validated by 5′ RACE using the HiScript-TS 5′/3′ RACE Kit (Vazyme, RA101), and the resulting amplified fragments were confirmed by Sanger sequencing. Gene-specific primers (GSPs) for 5′ RACE were designed using the Vazyme GSP primer design website (http://appbi.vazyme.com:8085/) and are listed in Supplementary Data 4.

## Reporting summary

Further information on research design is available in the Nature Portfolio Reporting Summary linked to this article.

## Data availability

All raw and processed datasets of ENCODE RAMPAGE, RNA-seq, and ChIP-seq are available at the ENCODE portal with the accessions listed in Supplementary Data 1. Source data are provided with this paper.

## Code availability

The source code of the TE-derived TSS identification pipeline can be accessed at GitHub (https://github.com/kepbod/rampage_te_tss)[64].

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

## Acknowledgements

This work was supported by grants from the National Key R&D Program of China (2022YFA1103200 to X.-O.Z.), the National Natural Science Foundation of China (32170553 to X.-O.Z.; 32000436 and 82170134 to H.W.), the Fundamental Research Funds for the Central Universities (22120250374 to X.-O.Z.), the Shanghai Action Plan for Science, Technology and Innovation (24JS2820200 to X.-O.Z.), the Shanghai Pilot Program for Basic Research (to X.-O.Z.), the Peak Disciplines (Type IV) of Institutions of Higher Learning in Shanghai (to X.-O.Z.), and the New Chongqing Youth Innovation Talent Project (CSTB2025YITP-QCRCX0040 to H.W.).

## Author contributions

X.-O.Z. conceived and designed the project. Y.Z. and X.-O.Z. developed the methodology and analyzed the results. Y.F. and H.W. conducted the experimental validation experiments. H.W. and X.-O.Z. acquired funding. X.-O.Z. wrote the original draft. Y.Z. and X.-O.Z. reviewed and edited the manuscript. All authors read and approved the final manuscript.

## Competing interests

The authors declare no competing interests.
