## [Transparent Peer Review file · Nature Communications]

Architectural and evolutionary features of TE-derived TSSs shape tissue-specific promoter activity in the human genome

Corresponding Author: Professor Xiao-Ou Zhang

A version of this paper was originally rejected for publication by Nature Communications, however that decision was reconsidered after appeal by the authors.

Version 0:

Reviewer comments:

Reviewer #1

(Remarks to the Author)

I co-reviewed this manuscript with an early-career researcher. This is part of the Nature Communications initiative to facilitate training in peer review. We first did a full review independently and after discussion I merged our reports with ChatGPT 5. We then both proof-checked the merged report, which is pasted below.

Charles Plessey

Summary

The manuscript by Zhang and Zhang presents a comprehensive atlas of more than 26,000 transposable element (TE)-derived transcription start sites (TSSs) across 115 human biosamples, generated using high-resolution RAMPAGE data. The authors show that TE-derived TSSs are crucial for tissue-specific gene expression and often serve as unique promoters for host genes, including some missing from GENCODE due to their TE origin. The study extends previous work (Gu et al., 2024; Miao et al., 2020) with new data and evolutionary analyses, offering a resource for future studies of promoter evolution and TE biology. Overall, this atlas has the potential to serve as a solid foundation for genome-wide studies of TE contributions to gene regulation and evolution.

Major Points

* High-confidence subset definition

- Many figures compare high-confidence versus all TE-derived TSSs against controls, yet they often look very similar. This raises questions about the added value of the high-confidence set. Consider contrasting high-confidence versus non-high-confidence directly.

- The thresholds for defining high-confidence TSSs (distance to other promoters, exon assignment fraction, gene length) are critical for readers to evaluate the subset's quality. Supplemental Figure S1 panels on these thresholds should be moved to the main text. Please also clarify whether the three thresholding dimensions are correlated.

* Clarification of results and methods

- In the sentence about "roughly half of TE-derived TSSs map within 1 kb", why is this only half? Is it due to RAMPAGE providing higher resolution than current annotations?

- Please elaborate in the Methods how the tissue-specificity score was calculated (Figure 2D).

* Evolutionary and functional implications

- Are there any differences in the function of target genes of TE-derived TSS that are exclusive in: hominids, hominids and primates, as compared to other mammals? If gene expression data is available, do orthologous genes exhibit different gene expression patterns due to these differences in promoters?
- Similarly, there is a difference in composition of TE subfamilies across different clades (Figure 3C), could this be linked to functional differences in target genes?

* Figures

- Consider separating Figure 3A into separate figures (or possibly removing the inset as can be represented by the “main” plot). Further, the numbers in the inset are different from the sum of the numbers at the “main” plot (e.g. SINE is 190 while Alu + MIR is 179). If the rest are other subfamilies, consider adding another boxplot, and thus the small inset could be omitted. The plot at the bottom can be separated into another plot as well (e.g. Figure 3B).
- Supplemental Figure S1C: color coding for cell lines is inconsistent with main-text figures, and the density curves have little vertical resolution. Consider alternative plots.
- Supplemental Figure S1D: the intron-exon diagram is misleading; log-log scaling might better illustrate distance thresholds.

Minor Points

- Use “RNA Pol II” instead of “POLR2A” (the latter refers to the gene rather than the protein).
- In the sentence “Next, we evaluated sequence patterns within the 1 kb downstream region of each TSS, in which canonical promoters display both enrichment of predicted 5’ splice-site motifs and depletion of polyadenylation signals”, the correct reference is Almada et al., Nature 2013.
- Add “exon” before “assignment fraction” in all instances for clarity.
- Check spelling, for example “FAMTOM” instead of “FANTOM.”
- Figure 1E: the “TSS” acronym near the poly(A) motif may be confused with “transcription stop site.” Please clarify.
- Figure 2A: the shape coding for sample type (tissue, primary cell, cell line) is hard to read; simplify or enhance.
- Figures 4C,E: the “ · Testis” labels are misaligned.
- The sentence starting with “To distinguish TE-derived TSSs from other TE-overlapping TSSs...” is unclear. Please rephrase.
- Some figure references seem mismatched (e.g., “Figure 1E,F” should be “1F,G”).
- “TE-derived” is used throughout, but “TE-initiated” appears twice — please clarify or standardize.
- Please add line numbers for ease of review.

(Remarks on code availability)

The GitHub repository (https://github.com/kepbod/rampage_te_tss) currently provides only a subset of the code with minimal documentation. At a minimum, please improve documentation. Ideally, include scripts used to generate figures presented in the manuscript.

Reviewer #2

(Remarks to the Author)

(Remarks on code availability)

Reviewer #3

(Remarks to the Author)

This manuscript presents a comprehensive analysis of transcription start sites (TSSs) associated with transposable

elements (TEs) in the human genome, primarily using RAMPAGE data. The topic is timely, relevant, and conceptually strong, as it addresses how repetitive sequences are exapted into regulatory functions. The work succeeds in assembling a large dataset and provides descriptive observations about tissue-restricted TSS usage and differences between younger and older TE subfamilies.

However, in its current form the study remains primarily descriptive and heavily dependent on a single detection pipeline. It does not incorporate alternative computational tools (e.g., ChimeraTE, LIONS, TEProf) or sample-matched validation data, which weakens confidence in the generality of the conclusions. Some figures are overly dense, and several claims could be more cautiously phrased to avoid overstating the impact. Overall, while the dataset has value, the manuscript requires substantial revisions to meet the standards of Nature Communications.

Specific points

1. Ambiguity in terminology: The use of “TE-overlapping TSSs” vs “TE-derived TSSs” is inconsistent and confusing. Clear definitions early in the text (or a glossary schematic) are needed.
2. Dataset metadata: Table S1 lacks essential sample information (tissue type, donor demographics, sequencing depth, QC metrics, accession links). Without these, readers cannot properly assess dataset quality or representativeness.
3. Methodological dependence on a single pipeline. The entire study relies on one RAMPAGE-based detection pipeline. Given the diversity of available TE-derived TSS detection methods (e.g. ChimeraTE, LIONS, TEProf), the conclusions would be more robust if cross-validated with at least one complementary approach. This would help address potential biases or artifacts unique to the chosen pipeline.
4. Overcrowded pipeline figures: Figures 1 and Sup. Fig. 1 attempt to show too many methodological details at once. Simplifying the pipeline schematic and moving QC checks to supplementary material would improve readability.
5. Validation strategy is limited. The validation analyses are restricted to GM12878 and K562, which are convenient but atypical immortalized cell lines. Their regulatory landscapes may not reflect primary tissues. The authors should either (a) incorporate validation in at least one additional primary tissue, or (b) more explicitly acknowledge this limitation.
6. Tissue specificity claims: The manuscript interprets rare, single-tissue occurrences as evidence of “extensive tissue specificity.” Replicate consistency and prevalence across donors should be assessed; otherwise, the claim should be moderated.
7. Contribution to gene expression: The heading “contribute substantially” is not well supported - TE-derived promoters account for ~5% of highly expressed genes, which is modest. A more cautious framing is needed.
8. Clarity of age classification. The “young” versus “old” TE categories need clearer operational definitions. For example, are age estimates based on divergence from consensus sequences, comparative genomics, or other metrics? Without transparent criteria, it is difficult to evaluate the conclusions about “preserved” promoter motifs.
9. Reliance on unmatched TF ChIP-seq: TF binding is inferred from datasets not matched to the RAMPAGE biosamples. Given context-specific binding, these analyses should be presented as suggestive rather than direct evidence.
10. Young vs. old TE promoters: The contrast between precise “young” and dispersed “old” promoters may be overstated. TE “age” criteria should be clarified, and the pattern reframed as a continuum rather than a strict dichotomy.

(Remarks on code availability)

Version 1:

Reviewer comments:

Reviewer #1

(Remarks to the Author)

The authors have adequately addressed my comments.

(Remarks on code availability)

Reviewer #2

(Remarks to the Author)

(Remarks on code availability)

Reviewer #3

(Remarks to the Author)

While the authors have satisfactorily addressed all my other concerns, I remain unconvinced by their response to this point: Although they clarify that their approach integrates RAMPAGE and RNA-seq data, the magnitude of discrepancy they report compared with TEProf2 (10- to 100-fold higher counts) suggests that their pipeline may be over-sensitive rather than uniquely accurate. Such inflation likely reflects permissive thresholds and residual mapping ambiguity within repetitive regions, where RAMPAGE read dispersion can generate artefactual TSS calls. Moreover, the cross-assay concordance reported with GRO-cap, Iso-Seq, and FANTOM CAGE represents general promoter overlap rather than true TE-initiated transcript confirmation, as these datasets share similar promoter-proximal biases. High replicate correlation ($r = 0.90^*$) demonstrates reproducibility of signal but not necessarily specificity or accuracy. Overall, the evidence provided does not convincingly rule out that the much larger RAMPAGE-based catalogs reflect pipeline-driven sensitivity and permissive filtering, rather than genuine biological expansion of TE-derived promoters; more stringent FDR estimation, independent TE-specific benchmarking, or orthogonal validation (e.g., targeted CAGE or reporter assays) would be required to substantiate these conclusions.

(Remarks on code availability)

Version 2:

Reviewer comments:

Reviewer #3

(Remarks to the Author)

The authors have now adequately addressed my comments.

(Remarks on code availability)

Point-by-Point Responses to Reviewers:

Reviewer #1 (Remarks to the Author)

I co-reviewed this manuscript with an early-career researcher. This is part of the Nature Communications initiative to facilitate training in peer review. We first did a full review independently and after discussion I merged our reports with ChatGPT 5. We then both proof-checked the merged report, which is pasted below.

Charles Plessey

Summary

The manuscript by Zhang and Zhang presents a comprehensive atlas of more than 26,000 transposable element (TE)-derived transcription start sites (TSSs) across 115 human biosamples, generated using high-resolution RAMPAGE data. The authors show that TE-derived TSSs are crucial for tissue-specific gene expression and often serve as unique promoters for host genes, including some missing from GENCODE due to their TE origin. The study extends previous work (Gu et al., 2024; Miao et al., 2020) with new data and evolutionary analyses, offering a resource for future studies of promoter evolution and TE biology. Overall, this atlas has the potential to serve as a solid foundation for genome-wide studies of TE contributions to gene regulation and evolution.

Response: We thank the reviewer for the positive and insightful summary of our study.

Major Points

* High-confidence subset definition

R1.1 Many figures compare high-confidence versus all TE-derived TSSs against controls, yet they often look very similar. This raises questions about the added value of the high-confidence set. Consider contrasting high-confidence versus non-high-confidence directly.

Response: We thank the reviewer for this helpful suggestion. In response to this comment, we directly compared high-confidence and non-high-confidence (RAMPAGE-only) TE-derived TSSs. The non-high-confidence set resembled high-confidence ones in several core promoter features: they exhibited a clear *Inr* motif (revised Figure 1C), characteristic downstream sequence context with enrichment of 5' splice-site motifs and depletion of polyadenylation signals (revised Figure 1E), and generate predominantly multi-exonic transcript isoforms (revised Supplemental Figures 2J,K). Although non-high-confidence TE-derived TSSs show reduced autonomous transcriptional activity in SuRE assays (revised Figure 1D) and less overlap with other TSS-mapping assays and curated promoter catalogs compared to high-confidence sites (revised Supplemental Figures S2D-H), they remain significantly more active and promoter-like than other TE-overlapping TSSs. These results indicate that the high-confidence set is useful when stringent support is required, while the non-high-confidence set captures additional bona fide TE-derived TSSs and helps extend the overall catalog. We have incorporated these comparisons into the revised manuscript.

R1.2 The thresholds for defining high-confidence TSSs (distance to other promoters, exon assignment fraction, gene length) are critical for readers to evaluate the subset's quality. Supplemental Figure S1 panels on these thresholds should be moved to the main text. Please also clarify whether the three thresholding dimensions are correlated.

Response: We thank the reviewer for this insightful comment. We would like to clarify that the classification of high-confidence versus non-high-confidence TE-derived TSSs is based solely on their distance to other promoters, whereas the exon-assignment fraction and genomic span are general parameters used to define TE-derived TSSs. Following your suggestion, we have moved the discussion of these thresholds to the main text for greater clarity.

By comparing these three thresholding dimensions among mRNA TSSs, autonomous TE TSSs, and annotated TE-derived TSSs, we show that these parameters provide clear separation and effectively support the identification of both TE-derived and high-confidence TE-derived TSSs (revised Supplemental Figures S1B-D). We further note that the exon-assignment fraction and genomic span are largely uncorrelated across samples (Spearman's ρ ranging from -0.20 to -0.10), indicating that they capture distinct transcript features rather than redundant information.

* Clarification of results and methods

R1.3 In the sentence about “roughly half of TE-derived TSSs map within 1 kb”, why is this only half? Is it due to RAMPAGE providing higher resolution than current annotations?

Response: We thank the reviewer for this question. The observation that roughly half of TE-derived TSSs map within 1 kb of annotated promoters reflects both the higher resolution of RAMPAGE and the presence of genuinely novel TE-driven initiation events. Because RAMPAGE captures precise transcription start sites at single-nucleotide resolution (Supplemental Figure S6A), some TE-derived TSSs show small positional shifts relative to existing annotations. Meanwhile, a substantial fraction of TE-derived TSSs occur in regions without any annotated promoters (Supplemental Figure S3A), consistent with exaptation of TEs as new transcription initiation sites. Thus, the “half within 1 kb” pattern arises from a combination of annotation differences and true novel promoter activity.

R1.4 Please elaborate in the Methods how the tissue-specificity score was calculated (Figure 2D).

Response: We thank the reviewer for the helpful suggestion. We have now added a detailed description of the calculation method for the tissue-specificity score in the revised Methods section under “Tissue specificity analysis.”

* Evolutionary and functional implications

R1.5 Are there any differences in the function of target genes of TE-derived TSS that are exclusive in: hominids, hominids and primates, as compared to other mammals? If gene expression data is available, do orthologous genes exhibit different gene expression patterns due to these differences in promoters?

Response: We thank the reviewer for this insightful comment. We analyzed lineage-restricted TE-derived TSSs and found that hominid-specific sites are preferentially associated with genes involved in neural, immune, and cardiac/muscle development, whereas primate-specific sites are enriched for immune, developmental, and metabolic processes. These enrichments suggest potential functional specialization linked to recent regulatory innovations. Although we have not yet performed a systematic cross-species expression comparison for orthologous genes, due to the limited availability of matched datasets, we

recognize this as an important direction for future work. The corresponding results and clarifications have been incorporated into the revised manuscript.

“To assess whether lineage- and subfamily-specific TE-derived TSSs correspond to functional divergence in host genes, we performed GO enrichment analysis. Genes associated with hominid-specific TE-derived TSSs were enriched for neural and immune programs and for cardiac or muscle-related development (Supplemental Figure S4E). Primate-specific TE-derived TSSs showed enrichment for immune, developmental, and metabolic processes (Supplemental Figure S4E).”

R1.6 Similarly, there is a difference in composition of TE subfamilies across different clades (Figure 3C), could this be linked to functional differences in target genes?

Response: We thank the reviewer for this constructive point. Indeed, TE subfamily composition differs markedly across evolutionary clades, and this divergence appears to parallel distinct functional enrichments of their target genes. Because ERV1 elements are strongly over-represented among primate-specific TE-derived TSSs and account for the majority of lineage-restricted promoters in this group, we further examined this subfamily as a representative example. We found that genes associated with primate-specific ERV1-derived TSSs were enriched for interferon signaling, cell adhesion, long-chain fatty acid metabolism, xenobiotic response, and transposable element silencing. These results suggest that evolutionary turnover of TE subfamilies has contributed to lineage-specific diversification of regulatory functions through recruitment of distinct promoter types. This addition has been incorporated into the revised manuscript.

“Because ERV1 elements are strongly over-represented among primate-specific TSSs, we further examined genes associated with primate-specific ERV1-derived TSSs and found enrichment for interferon signaling, cell–cell adhesion, long-chain fatty acid metabolism, xenobiotic response, and transposable element silencing (Supplemental Figure S4F).”

* Figures

R1.7 Consider separating Figure 3A into separate figures (or possibly removing the inset as can be represented by the “main” plot). Further, the numbers in the inset are different from the sum of the numbers at the “main” plot (e.g. SINE is 190 while Alu + MIR is 179). If the rest are other subfamilies, consider adding another boxplot, and thus the small inset could be omitted. The plot at the bottom can be separated into another plot as well (e.g. Figure 3B).

Response: We thank the reviewer for the constructive suggestion. In the revised version, we have incorporated the other subfamilies into the main panel and removed the small inset accordingly. Additionally, the plot previously shown at the bottom has been moved to a new panel (revised Figure 3B) to improve clarity.

R1.8 Supplemental Figure S1C: color coding for cell lines is inconsistent with main-text figures, and the density curves have little vertical resolution. Consider alternative plots.

Response: We thank the reviewer for the suggestion. In the revised Supplemental Figure S1C, we have unified the color scheme to match the main-text figures: red denotes host genes with TE-derived TSSs, and gray denotes expressed genes. Additionally, we replaced the density curves with boxplots to improve vertical resolution and visual clarity.

R1.9 Supplemental Figure S1D: the intron-exon diagram is misleading; log-log scaling might better illustrate distance thresholds.

Response: We thank the reviewer for the suggestion. In the revised Supplemental Figure S1D, we removed the intron-exon diagram and applied log–log scaling to better illustrate distance distributions and thresholds.

Minor Points

R1.10 Use “RNA Pol II” instead of “POLR2A” (the latter refers to the gene rather than the protein).

Response: *We thank the reviewer for this suggestion. We have replaced all instances of “POLR2A” with “RNA Pol II” throughout the manuscript to accurately refer to the protein.*

R1.11 In the sentence "Next, we evaluated sequence patterns within the 1 kb downstream region of each TSS, in which canonical promoters display both enrichment of predicted 5' splice-site motifs and depletion of polyadenylation signals", the correct reference is Almada et al., Nature 2013.

Response: *We thank the reviewer for the correction. We have updated the reference to Almada et al., Nature, 2013 in the revised manuscript.*

R1.12 Add “exon” before “assignment fraction” in all instances for clarity.

Response: *We thank the reviewer for the suggestion. We have added “exon” before “assignment fraction” throughout the manuscript for clarity.*

R1.13 Check spelling, for example “FAMTOM” instead of “FANTOM.”

Response: *We thank the reviewer for pointing this out. We have corrected the spelling error (“FAMTOM” to “FANTOM”) in the revised manuscript.*

R1.14 Figure 1E: the “TSS” acronym near the poly(A) motif may be confused with “transcription stop site.” Please clarify.

Response: *We thank the reviewer for this helpful comment. We have replaced “TSS” with “transcription start site” in Figure 1E (revised as Figure 1F) to avoid any potential confusion.*

R1.15 Figure 2A: the shape coding for sample type (tissue, primary cell, cell line) is hard to read; simplify or enhance.

Response: *We thank the reviewer for the helpful suggestion. In the revised Figure 2A, we use solid circles to denote tissue samples, crosses to denote primary cells, and open circles to denote cell lines. To improve readability, the symbols have been enlarged and their spacing adjusted to enhance visual distinction.*

R1.16 Figures 4C,E: the “ · Testis” labels are misaligned.

Response: *We thank the reviewer for pointing this out. The “Testis” labels have been removed from Figures 4C and 4E and are now included in the figure legend to avoid misalignment and improve clarity.*

R1.17 The sentence starting with “To distinguish TE-derived TSSs from other TE-overlapping TSSs...” is unclear. Please rephrase.

Response: *We thank the reviewer for the comment. The unclear sentence has been removed, and the related discussion of thresholds has been moved to the main text for greater clarity.*

R1.18 Some figure references seem mismatched (e.g., “Figure 1E,F” should be “1F,G”).

Response: We thank the reviewer for pointing out this mismatch. In the revised manuscript, the relevant panels have been moved to the supplementary figures, and the references have been updated accordingly (now referred to as Supplemental Figures S2J,K).

R1.19 “TE-derived” is used throughout, but “TE-initiated” appears twice — please clarify or standardize.

Response: We thank the reviewer for pointing this out. We have standardized the terminology throughout the manuscript by replacing the two occurrences of “TE-initiated” with “TE-derived” to maintain consistency.

R1.20 Please add line numbers for ease of review.

Response: We appreciate the reviewer’s suggestion. Line numbers have been added throughout the revised manuscript to facilitate review.

(Remarks on code availability)

R1.21 The GitHub repository (https://github.com/kepbod/rampage_te_tss) currently provides only a subset of the code with minimal documentation. At a minimum, please improve documentation. Ideally, include scripts used to generate figures presented in the manuscript.

Response: We thank the reviewer for this suggestion. We have updated the GitHub repository to improve documentation, including detailed instructions for running the provided code. In addition, we have now included the scripts used to generate the figures presented in the manuscript, enabling full reproducibility of our analyses.

Reviewer #2 (Remarks to the Author)

Response: We thank the reviewer for co-reviewing this manuscript.

(Remarks on code availability)

Reviewer #3 (Remarks to the Author)

This manuscript presents a comprehensive analysis of transcription start sites (TSSs) associated with transposable elements (TEs) in the human genome, primarily using RAMPAGE data. The topic is timely, relevant, and conceptually strong, as it addresses how repetitive sequences are exapted into regulatory functions. The work succeeds in assembling a large dataset and provides descriptive observations about tissue-restricted TSS usage and differences between younger and older TE subfamilies.

However, in its current form the study remains primarily descriptive and heavily dependent on a single detection pipeline. It does not incorporate alternative computational tools (e.g., ChimeraTE, LIONS, TEProf) or sample-matched validation data, which weakens confidence in the generality of the conclusions. Some figures are overly dense, and several claims could be more cautiously phrased to avoid overstating the impact. Overall, while the dataset has value, the manuscript requires substantial revisions to meet the standards of Nature Communications.

Response: We thank the reviewer for the overall assessment and constructive feedback. We appreciate the recognition of the timeliness and conceptual significance of our study, as well as the value of the large TE-derived TSS dataset. In response to the concerns raised, we have made several substantial revisions:

1. Methodological robustness: We clarified our RAMPAGE-integrated pipeline, incorporated matched RNA-seq support for transcript validation, and performed cross-validation with TEProf2 on representative samples to demonstrate reproducibility and sensitivity.

2. Terminology and clarity: Definitions of “TE-overlapping” versus “TE-derived” TSSs are now consistent, and we clarified TE evolutionary age and promoter activity trends as a continuous gradient rather than a strict dichotomy.

3. Figure clarity: Overcrowded pipeline figures were simplified and quality-control steps moved to the supplementary material to improve readability.

4. Tone moderation: Several claims, including TF binding interpretation and the contribution of TE-derived TSSs to gene expression, have been carefully toned down to avoid overstating the impact.

We hope that these revisions address the reviewer’s concerns regarding descriptive emphasis, pipeline dependence, and figure readability, while retaining the novel insights and value of our TE-derived TSS atlas across human tissues.

Specific points

R3.1 Ambiguity in terminology: The use of “TE-overlapping TSSs” vs “TE-derived TSSs” is inconsistent and confusing. Clear definitions early in the text (or a glossary schematic) are needed.

Response: We thank the reviewer for this comment. We have clarified the terminology throughout the manuscript. Specifically, “TE-overlapping TSSs” now consistently refers to all RAMPAGE peaks that overlap annotated transposable elements, whereas “TE-derived TSSs” refers to a subset of TE-overlapping TSSs that meet additional criteria—namely, RAMPAGE read pairs linking the peak to exonic regions of annotated or de novo transcripts, genomic span > 1 kb, and exon-assignment fraction ≥ 0.5 . These definitions are summarized in a schematic in Supplemental Figure S1A and described explicitly in the Methods section as list below.

“Peaks overlapping transposon elements (hg38 rmsk.txt downloaded from UCSC) were defined as TE-overlapping TSSs.

Candidate TE-derived TSSs were selected if (i) paired RAMPAGE reads linked the peak to exonic regions of annotated or de novo transcripts, (ii) the genomic span of the read pair was > 1 kb, and (iii) the exon-assignment fraction ≥ 0.5 (Supplemental Figures S1B,C; Supplemental Table S2).”

R3.2 Dataset metadata: Table S1 lacks essential sample information (tissue type, donor demographics, sequencing depth, QC metrics, accession links). Without these, readers cannot properly assess dataset quality or representativeness.

Response: We thank the reviewer for this valuable suggestion. We have updated Supplemental Table S1 to include comprehensive metadata for all 115 RAMPAGE biosamples, including tissue type, donor age, sequencing depth, number of uniquely mapped reads, mapping ratio, and direct ENCODE accession links. These additions ensure full transparency and allow readers to readily assess the quality, representativeness, and provenance of all datasets used in this study.

R3.3 Methodological dependence on a single pipeline. The entire study relies on one RAMPAGE-based detection pipeline. Given the diversity of available TE-derived TSS detection methods (e.g. ChimeraTE, LIONS, TEProf), the

conclusions would be more robust if cross-validated with at least one complementary approach. This would help address potential biases or artifacts unique to the chosen pipeline.

Response: *We thank the reviewer for raising this important point. Our pipeline is not solely RAMPAGE-based and also integrates matched RNA-seq data to facilitate the identification of TE-derived TSSs. We used RNA-seq data to assemble transcripts with StringTie and required that RAMPAGE read pairs link each candidate TSS to downstream exonic regions of assembled transcripts, with an exon-assignment fraction of at least 0.5. This design leverages the single-nucleotide precision of RAMPAGE while using RNA-seq as supportive evidence for transcript structure.*

In contrast, other available TE-derived TSS detection methods rely only on RNA-seq data. They perform de novo transcript assembly and read assignment to annotate putative TE-derived TSSs, and therefore cannot directly capture 5' -capped transcripts or achieve nucleotide-level TSS resolution.

To evaluate robustness and potential pipeline-specific biases, we applied TEProf2, which also uses StringTie assemblies that have been reported to outperform Cufflinks used by ChimeraTE and LIONS, to five representative samples (GM12878, K562, heart, liver, and testis). TEProf2 identified far fewer candidate sites than our RAMPAGE-integrated pipeline, yielding 30, 50, 34, 38, and 122 sites compared with 961, 861, 554, 1150, and 5052 sites, respectively. Most TEProf2 calls were recovered by our pipeline, with overlap counts of 24, 36, 30, 26, and 99, corresponding to 80.0%, 72.0%, 88.2%, 68.4%, and 81.1% of TEProf2 calls in each sample. In contrast, these overlapping sites represent only 2.5%, 4.2%, 5.4%, 2.3%, and 2.0% of our larger RAMPAGE catalogs. These results indicate that RNA-seq-only methods recover only a small subset of TE-derived promoters also detected by our approach, whereas the integration of RAMPAGE data markedly enhances promoter discovery sensitivity

We further assessed reproducibility across biological replicates and found that in 50 biosamples with biological replicates, the median Pearson correlation of TE-derived TSS signal between replicates was 0.90, demonstrating high reproducibility of our calls.

Together, these analyses demonstrate that TEProf2, which relies on RNA-seq data, captures a subset of sites largely encompassed by our RAMPAGE calls, that differences in call numbers primarily reflect assay sensitivity rather than artifacts, and that our RAMPAGE-integrated pipeline is both robust and reproducible.

R3.4 Overcrowded pipeline figures: Figures 1 and Sup. Fig. 1 attempt to show too many methodological details at once. Simplifying the pipeline schematic and moving QC checks to supplementary material would improve readability.

Response: *We thank the reviewer for this constructive suggestion. To address the concern of overcrowding, we have simplified the main pipeline schematic in Figure 1A to display only the essential steps for identifying TE-derived TSSs, while detailed quality control procedures are now presented in Supplemental Figure S1A. Additionally, transcript structure validations of TE-derived TSSs, previously shown in Figures 1F–G, have been moved to Supplemental Figures S2J,K. These revisions maintain full methodological transparency while ensuring that the main figure conveys the workflow clearly and is easier to interpret.*

R3.5 Validation strategy is limited. The validation analyses are restricted to GM12878 and K562, which are convenient but atypical immortalized cell lines. Their regulatory landscapes may not reflect primary tissues. The authors should either (a) incorporate validation in at least one additional primary tissue, or (b) more explicitly acknowledge this limitation.

Response: *We thank the reviewer for this valuable comment. In addition to GM12878 and K562, we have now included validation analyses in three primary tissues (heart, liver, and testis) to strengthen the generality of our findings (revised Supplemental Figure 1 and 2).*

R3.6 Tissue specificity claims: The manuscript interprets rare, single-tissue occurrences as evidence of “extensive

tissue specificity.” Replicate consistency and prevalence across donors should be assessed; otherwise, the claim should be moderated.

Response: We thank the reviewer for this helpful comment. We have assessed replicate consistency across 50 biosamples with biological replicates, and the median Pearson correlation coefficient of TE-derived TSS signals was 0.90 (revised Supplemental Figure S2C), confirming the strong reproducibility of our RAMPAGE calls. This high consistency supports the robustness of our tissue-specificity analysis.

R3.7 Contribution to gene expression: The heading “contribute substantially” is not well supported - TE-derived promoters account for ~5% of highly expressed genes, which is modest. A more cautious framing is needed.

Response: We appreciate the reviewer’s careful reading. We fully acknowledge that TE-derived TSSs occur in a relatively small fraction of highly expressed genes (~5%), and we have clarified this point in the revised manuscript. At the same time, our analyses demonstrate that when TE-derived TSSs are present, they can make a substantial contribution at the level of individual genes. Specifically, more than one quarter of genes with TE-derived TSSs derive over half of their promoter activity from these sites, and roughly half exceed a ten percent usage threshold (Figure 4D–F; Supplemental Figure S7). Thus, while TE-derived TSSs are not globally prevalent across all highly expressed genes, they frequently act as dominant promoters for a notable subset of genes. We have therefore (i) softened the Results heading to “TE-derived TSSs are enriched for narrow-peak promoter architectures and contribute to gene expression”, and (ii) added sentences in the Discussion clarifying the distinction between global prevalence and per-gene contribution so that readers can correctly interpret both aspects.

“Although TE-derived TSSs appear in only around five percent of highly expressed genes, their contribution at the level of individual genes can still be substantial. Approximately one quarter of genes with TE-derived TSSs derive more than half of their promoter activity from these sites, and roughly half show at least ten percent usage, with over thirty percent of TE-derived TSSs functioning as the predominant or unique promoter. These findings indicate that, despite their limited genome-wide prevalence, TE-derived promoters often act as dominant regulatory elements for a considerable subset of genes, thereby contributing to transcriptomic diversity and regulatory innovation in specific genomic contexts.”

R3.8 Clarity of age classification. The “young” versus “old” TE categories need clearer operational definitions. For example, are age estimates based on divergence from consensus sequences, comparative genomics, or other metrics? Without transparent criteria, it is difficult to evaluate the conclusions about “preserved” promoter motifs.

Response: We thank the reviewer for this comment. To clarify, we define the evolutionary age of TE subfamilies based on the percent sequence divergence of individual TE copies from their consensus sequences, which serves as a continuous proxy for time since insertion. Lower sequence divergence corresponds to relatively younger TE insertions, whereas higher divergence indicates older elements. For clarity, we have added this description to the Results section in the revised manuscript.

“To estimate evolutionary age, we used the percent divergence of each TE copy from its consensus sequence as a proxy for time since insertion.”

R3.9 Reliance on unmatched TF ChIP-seq: TF binding is inferred from datasets not matched to the RAMPAGE biosamples. Given context-specific binding, these analyses should be presented as suggestive rather than direct evidence.

Response: We thank the reviewer for this comment. We have toned down the language regarding TF binding in the main text to reflect that these observations are suggestive rather than direct evidence. The revised paragraph emphasizes enrichment patterns without over-interpreting them.

“Finally, to explore which transcription factors (TFs) are preferentially associated with TE-derived TSSs, we overlapped them with ChIP-seq peaks from 694 TFs curated by the ENCODE Consortium³⁶ (Supplemental Figure S5A). TE-derived TSSs tend to be bound by a broader repertoire of TFs compared to promoter-proximal TEs (Chi-squared test, $p < 2.2 \times 10^{-16}$; Figure 3H). Several general TFs—including POLR2A, TBP, TAFs, and YY1—show enrichment across TE-derived TSSs from diverse subfamilies (Figure 3H). Aggregate ChIP-seq signal profiles in matched biosample further support stronger binding of these TFs at TE-derived TSSs relative to promoter-proximal TEs (Supplemental Figure S5B). In contrast, several TFs exhibited subfamily-specific patterns suggestive of preferential TF association, supported by both ChIP-seq signal and motif enrichment (Supplemental Figure S5C). For instance, ETS1 and ETV5, which recognize canonical ETS motifs³⁷, were preferentially enriched at MIR-derived TSSs. Similarly, NFYA and NFYB, which bind CCAAT boxes³⁸, showed selective enrichment at ERV1- and ERVL-MaLR-derived TSSs. USF1 and USF2, which recognize E-box motifs³⁹, were enriched at ERVL and ERVL-MaLR TSSs. Together, these results suggest that specific TE subfamilies may retain recognizable TF motifs that could facilitate subfamily-specific regulatory activity. Notably, TF binding patterns at TE-derived TSSs appear distinct from those governing autonomous TE transcription. For example, we previously showed that AP-1 family members such as FOS and JUN preferentially bind Pol III-transcribed Alu elements to promote their expression²⁴. However, this enrichment was absent from Alu-derived TSSs (Supplemental Figure S5D), which is suggestive of a transition from internal TE-driven transcription to host-mediated co-option of TE fragments as regulatory modules.”

R3.10 Young vs. old TE promoters: The contrast between precise “young” and dispersed “old” promoters may be overstated. TE “age” criteria should be clarified, and the pattern reframed as a continuum rather than a strict dichotomy.

Response: We thank the reviewer for this suggestion. We have clarified that TE “age” is defined continuously based on sequence divergence from consensus sequences, and reframed the differences in promoter precision and strength as a gradient rather than a strict dichotomy. Younger, low-divergence TEs retain promoter motifs and drive tightly clustered, high-activity TSSs, whereas more diverged elements show broader TSS placement and reduced intrinsic activity. Corresponding revisions have been made in the Abstract, Results, and Discussion sections.

“Phylogenetic analyses reveal a continuous gradient in promoter strength and transcriptional precision across TE subfamilies, with evolutionarily younger TEs retaining intrinsic promoter motifs that drive focused and robust transcription, whereas older, more diverged elements exhibit broader TSS usage and lower intrinsic activity.

Together, these findings indicate that promoter precision and activity form a continuous evolutionary gradient that parallels TE sequence divergence. Elements with low divergence preserve their intrinsic promoter architecture and drive strong, well-defined initiation, while those that have accumulated substitutions and deletions exhibit dispersed transcription initiation and increasing dependence on host regulatory environments.

Our evolutionary analysis reveals a continuous gradient in the transcriptional precision and strength of TE-derived TSSs that parallels sequence divergence and phylogenetic age. Younger TE subfamilies restricted to primates and hominids retain intact core promoter motifs and drive tightly clustered initiation at conserved positions, resulting in precise and robust transcription. In contrast, older lineages such as MIR and L2 exhibit higher sequence divergence, more dispersed TSS placement, weaker intrinsic strength, and partial loss of structural integrity, reflecting gradual promoter decay through sequence erosion and selection^{18,33}. Previous reports have demonstrated that primate-specific TEs often harbor lineage-restricted transcription factor binding sites active during early development^{51,52}, consistent with the longer retention of functional motifs in younger elements. The strong correlation between TSS clustering and promoter strength across Alu, L1, and ERVL-MaLR families further supports a model in which promoter exaptation follows a continuous evolutionary trajectory, shaped by the preservation of intrinsic sequence features in younger TEs and the surrounding genomic context in older insertions.”

(Remarks on code availability)

Point-by-Point Responses to Reviewers:

Reviewer #1 (Remarks to the Author):

The authors have adequately addressed my comments.

Response: We thank the reviewer for their encouraging feedback and are pleased that the revisions have adequately addressed the concerns.

Reviewer #2 (Remarks to the Author):

Response: We thank the reviewer for co-reviewing this manuscript.

Reviewer #3 (Remarks to the Author):

While the authors have satisfactorily addressed all my other concerns, I remain unconvinced by their response to this point: Although they clarify that their approach integrates RAMPAGE and RNA-seq data, the magnitude of discrepancy they report compared with TEProf2 (10- to 100-fold higher counts) suggests that their pipeline may be over-sensitive rather than uniquely accurate. Such inflation likely reflects permissive thresholds and residual mapping ambiguity within repetitive regions, where RAMPAGE read dispersion can generate artefactual TSS calls. Moreover, the cross-assay concordance reported with GRO-cap, Iso-Seq, and FANTOM CAGE represents general promoter overlap rather than true TE-initiated transcript confirmation, as these datasets share similar promoter-proximal biases. High replicate correlation ($r = 0.90^*$) demonstrates reproducibility of signal but not necessarily specificity or accuracy. Overall, the evidence provided does not convincingly rule out that the much larger RAMPAGE-based catalogs reflect pipeline-driven sensitivity and permissive filtering, rather than genuine biological expansion of TE-derived promoters; more stringent FDR estimation, independent TE-specific benchmarking, or orthogonal validation (e.g., targeted CAGE or reporter assays) would be required to substantiate these conclusions.

Response: We thank the reviewer for raising this important point regarding potential over-sensitivity of our TE-derived TSS pipeline in repetitive regions. We would like to emphasize that multiple independent lines of evidence—including analyses and experiments newly added in this revision—support the accuracy and biological relevance of our TE-derived TSS annotations, rather than reflecting artefactual calls:

1. Stringent read mapping and peak calling:

- **We exclusively used uniquely mapped RAMPAGE read pairs to call TE-derived TSSs and applied our previously developed entropy (E) metric to distinguish true TSS peaks from background noise in repetitive loci, which has been validated in our previous work (Zhang et al., *Genome Research*, 2019).**
- **Notably, RAMPAGE has been independently shown to accurately identify transcription start sites of transposable elements (Batut et al., *Genome Research*, 2013; Zhang et al., *Genome Research*, 2019) as well as protein-coding genes (Moore et al., *Genome Research*, 2021).**

2. Evaluation of TE-derived TSS precision:

- **Overlapping our annotated TE-derived TSSs with RAMPAGE peaks from K562 and GM12878 showed that nearly all peaks fall within ± 20 nt of their corresponding TE-derived TSSs (Supplemental Figure S6A; permutation test P -values $< 1 \times 10^{-4}$).**
- **Furthermore, newly added analyses in this revision (new Supplemental Figure S6B; see below) show that the variability of RAMPAGE peak summits at TE-derived TSSs is comparable to that of mRNA TSSs and significantly lower than for other TE-overlapping TSSs, indicating precise and reproducible localization rather than dispersed or ambiguous initiation within repetitive regions.**

- **In addition, we note that the majority of high-confidence TE-derived TSSs identified by our pipeline (~80%) overlap previously annotated transcription start sites (Supplemental Figure S3A), indicating that the expanded catalog largely reflects systematic recovery of bona fide promoters rather than artefactual inflation driven by permissive thresholds.**

3. Intrinsic promoter activity:

- **TE-derived TSSs exhibit significantly higher intrinsic transcriptional activity than other TE-overlapping TSSs in SuRE assays (Figure 1D).**
- **In addition, luciferase reporter assays performed in this revision using four randomly selected TE-derived TSSs demonstrate robust, strand-specific promoter activity, with all tested TSSs driving luciferase expression significantly above the empty vector control and exceeding the activity of two previously reported TE-derived TSSs (new Figure 1E and revised Supplemental Figure S2A; see below).**

4. Evidence for genuine TE-initiated transcripts:

- **The 1 kb downstream regions of TE-derived TSSs recapitulate canonical promoter features, including enrichment of 5' splice-site motifs and depletion of polyadenylation signals (Figure 1F).**
- **Newly added in this revision, a significantly higher proportion of TE-derived TSSs are supported by splice-junction-spanning RAMPAGE read pairs, compared with other TE-overlapping TSSs (new Figure 1G; see below).**

- **Independent de novo assembly of RAMPAGE reads confirms that TE-derived TSSs generate multi-exonic transcripts, whereas other TE-overlapping sites predominantly yield short, mono-exonic assemblies (Supplemental Figures S2C–D).**
- **Finally, also newly added in this revision, 5' RACE and Sanger sequencing of four randomly selected, previously unannotated TE-derived TSSs in K562 cells confirmed the transcription initiation events identified by our pipeline (new Figure 1H; revised Supplemental Figure S2E; see below).**

Collectively, these newly added analyses and experimental validations demonstrate that our TE-derived TSS catalog is accurate, reproducible, and biologically meaningful, and that the expanded number of TE-derived TSSs reflects genuine promoter activity rather than pipeline artefacts.